# SPICE: Submodular Penalized Information–Conflict Selection for Efficient Large Language Model Training

**Powei Chang**[*]  **Jinpeng Zhang**[*]  **Bowen Chen**  **Chenyu Wang**  **Chenlu Guo**  **Yixing Zhang**
**Yukang Gao**  **JianXiang Xiang**  **Yue Gao**  **Chaoqun Sun**  **Yiyi Chen**  **Dongying Kong**[†]

**Bilibili Inc.**   [*] **Equal contribution**   [†] **Corresponding author**
Correspondence to: {zhangbowei01,zhangjinpeng01,kongdongying}@bilibili.com

## Abstract

Information-based data selection for instruction tuning is compelling: maximizing the log-determinant of the Fisher information yields a monotone submodular objective, enabling greedy algorithms to achieve a $(1 - 1/e)$ approximation under a cardinality budget. In practice, however, we identify alleviating gradient conflicts, misalignment between per-sample gradients, is a key factor that slows down the decay of marginal log-determinant information gains, thereby preventing significant loss of information. We formalize this via an $\varepsilon$-decomposition that quantifies the deviation from ideal submodularity as a function of conflict statistics, yielding data-dependent approximation factors that tighten as conflicts diminish. Guided by this analysis, we propose SPICE, a conflict-aware selector that maximizes information while penalizing misalignment, and that supports early stopping and proxy models for efficiency. Empirically, SPICE selects subsets with higher log-determinant information than original criteria, and these informational gains translate into performance improvements: across 8 benchmarks with LLaMA2-7B and Qwen2-7B, SPICE uses only 10% of the data, yet matches or exceeds 6 methods including full-data tuning. This achieves performance improvements with substantially lower training cost. Code is available at code.

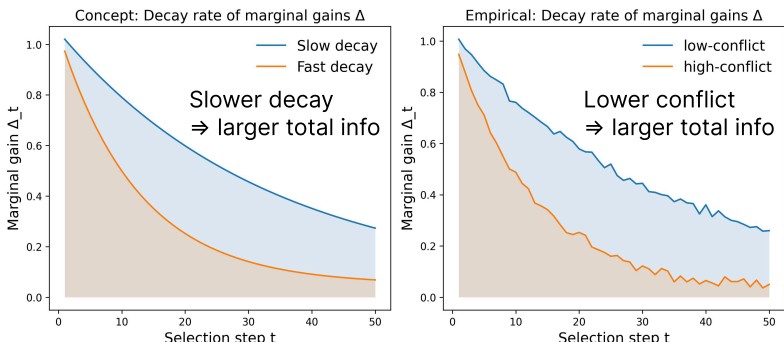

Figure 1: **(a) Concept.** At step $t$, the marginal information gain $\Delta_t$ is the incremental increase of Fisher Information utility when adding one sample under the current set $S$. **Slower decay** of $\Delta_t$ yields larger cumulative information under the same budget $k$. **(b) Empirical.** A low-conflict selection (conflict measured by negative cosine alignment to the mean gradient) exhibits slower decay and thus higher information utility at equal budgets.

## 1 Introduction

The remarkable capabilities of large language models (LLMs) have fundamentally transformed natural language processing. However, their performance critically depends on the quality and composition of instruction-tuning data (Albalak et al., 2024). Instruction tuning has emerged as an effective paradigm for improving both performance and alignment by fine-tuning with instruction-response pairs (Chang et al., 2023). Surprisingly, increasing the amount of training data does not always yield

better performance (Albalak et al., 2024; Zhou et al., 2023b; Li et al., 2024b). Recent empirical studies reveal a striking phenomenon: in instruction tuning, training on only 10–20% of the total data can outperform using the full dataset (Xia et al., 2024b). This observation raises a fundamental question: *how can we match or surpass full-data tuning performance with less data?*

Data selection has emerged as a promising solution to the high cost of large-scale training. Recent advances demonstrate notable gains: LESS (Xia et al., 2024a) attains 90% of full-data performance with only 5% of training samples, while SelectIT (Liu et al., 2025) achieves comparable results with 10–20%. Among these approaches, gradient-based selection methods show great promise. In particular, approaches based on the log-determinant of the Fisher Information Matrix (log-det FIM) offer a principled formulation and certain theoretical guarantees via submodularity (Fisher, 1922; Deb et al., 2025). However, these methods suffer from heavy computational overhead (e.g., sometimes exceeding 100 GPU-hours) and offer only partial theoretical explanations for their empirical behavior, leaving a noticeable gap between theory and practice.

In information-based selection, this gap is particularly evident in greedy selection with FIM as the objective (Deb et al., 2025). While theoretical guarantees suggest consistently near-optimal performance, practitioners observe that greedy selection often deteriorates far more rapidly than predicted. In fact, some selections perform substantially worse than the bound implies, and the discrepancy becomes particularly pronounced once the selected subset exceeds a certain threshold. At this point, marginal (information) gain, i.e., the incremental increase in the log-det FIM when adding one sample under a current set, collapse rapidly despite theoretical assurances of gradual diminishing returns. When marginal information gains diminish more slowly, they yield larger total information under a cardinality budget. Such observations naturally prompt a central question: *what causes this theory–practice gap, and how can it be bridged?*

Our key insight addresses the missing factor: the decay rate of marginal information gains. The log-det FIM is indeed submodular (Lovász, 1983), ensuring diminishing returns. However, its marginal information gains do not decrease uniformly. Instead, the decay rate varies substantially depending on interactions among sample gradients. We demonstrate that this variability is governed by **gradient conflicts**. These occur when gradients of different samples are poorly aligned (Liu et al., 2024a). Importantly, we go beyond prior gradient alignment work by establishing a direct connection between conflicts and submodular data selection theory, rather than focusing solely on training dynamics.

We provide a novel theoretical framework that quantitatively links gradient conflicts to marginal information gains decay. Our analysis splits marginal information gains into a submodular baseline and a perturbation term. We prove that the magnitude of this perturbation, which determines how quickly the decay of marginal information gains, is bounded by the sum of squared gradient inner products. When gradients are highly misaligned, this perturbation grows, accelerating information gains decay. This result explains why greedy approximation guarantees weaken in practice: *conflicts accelerate the erosion of marginal information gains*. Importantly, this does not challenge the submodularity of the log-det FIM, but rather provides a deeper understanding of its practical behavior.

Building on this theoretical framework, we design a conflict-aware greedy selection algorithm: **SPICE** (**S**ubmodular **P**enalized **I**nformation–**C**onflict s**E**lection). SPICE maximizes the log-det FIM objective while adaptively penalizing gradient conflicts. It incorporates a data-driven early stopping rule that selects an appropriate subset size as marginal information gains decay. This design preserves computational efficiency while yielding substantially higher-quality subsets. Finally, we conduct extensive experiments to validate the effectiveness of our approach.

Our contributions can be summarized as follows:

(1) **Theoretical Framework.** To our knowledge, we establish the first quantitative $\varepsilon$-analysis linking gradient conflicts to the decay of marginal FIM gains. We provide formal bounds suggesting that the effective submodularity curvature, which governs greedy approximation quality, is directly controlled by gradient conflict intensity, bridging the theory-practice gap for subsequent optimization. These theoretical insights are also validated empirically, demonstrating their practical effectiveness.

(2) **Conflict-Aware Algorithm.** Guided by this theory, we propose SPICE, a greedy selection method that adaptively penalizes conflicts rather than discarding informative samples. SPICE

achieves efficient selection complexity of $\mathcal{O}(k|D|d)$ and demonstrates consistent improvements over state-of-the-art baselines across 8 benchmarks while training on only 10% of the data with a total cost of 20 GPU-hours. Overall, we demonstrate empirically that such larger information translates into practical gains on downstream tasks.

Beyond practical gains, our work highlights a central insight: **gradient conflicts are a key factor underlying the accelerated diminishing returns in data selection**. By making the decay rate of marginal information gains explicit and tractable, we offer both a sharper theoretical understanding and a practical tool for constructing compact, high-quality training sets. This framework opens new directions for understanding and improving data efficiency in large-scale training, even billion-scale LLMs, with potential applications spanning computer vision, reinforcement learning, and multimodal learning.

## 2 THEORETICAL GUARANTEES

Gradient-based data selection methods frequently fall short of their theoretical guarantees. Existing methods like FisherSFT (Deb et al., 2025) rely on greedy maximization of the FIM, which enjoys strong submodularity guarantees. Yet practitioners observe performance degradation that deviates from the smooth diminishing returns that theory suggests. Our key insight is that sample interactions lead to non-uniform decay in marginal gains, accelerating performance degradation beyond classical bounds. We formalize this through a novel $\varepsilon$-decomposition that explains the theory-practice gap and provides data-dependent bounds for better selection algorithms.

### 2.1 PRELIMINARIES

We consider data selection from $\mathcal{D} = \{(x_i, y_i)\}_{i=1}^{N}$ instruction-response pairs, aiming to find a subset $S \subseteq \mathcal{D}$ with $|S| \leq k$ that maximizes learning efficiency (e.g., downstream performance).

**Definition 1** (Fisher Information Utility). *For each sample $i$, let $g_i = \nabla_\theta \ell((x_i, y_i); \theta)$ denote the gradient vector. The empirical Fisher information matrix (FIM) is $\mathbf{F}_S = \sum_{i \in S} g_i g_i^\top$, and the utility function is:*

$$F(S) = \log \det(\mathbf{I} + \alpha \mathbf{F}_S), \tag{1}$$

*where $\alpha > 0$ ensures positive definiteness.*

For computational efficiency in large models, we also consider the AdaFisher variant (Gomes et al., 2025) that reduces complexity from $\mathcal{O}(d^2)$ to $\mathcal{O}(d)$ (details in Appendix A), while preserving the same utility structure.

**Definition 2** (Submodular Function). *A set function $F : 2^{\mathcal{D}} \to \mathbb{R}$ is submodular if for all $A \subseteq B \subseteq \mathcal{D}$ and $x \in \mathcal{D} \setminus B$, the marginal gain $\Delta_x(S) \triangleq F(S \cup \{x\}) - F(S)$ satisfies:*

$$\Delta_x(A) \geq \Delta_x(B). \tag{2}$$

This diminishing returns property ensures that greedy algorithms achieve a $(1 - 1/e)$ approximation to the optimal solution in Theorem 2 (Nemhauser et al., 1978).

### 2.2 $\varepsilon$-DECOMPOSITION: BEYOND STANDARD SUBMODULARITY

While the Fisher utility $F(S) = \log \det(\mathbf{I} + \alpha \mathbf{F}_S)$ is indeed submodular (proof in Appendix B), this standard result alone cannot explain why greedy selection often deteriorates so rapidly in practice. The limitation is that submodularity only guarantees diminishing returns, but says nothing about the *rate* of this decay. Our key insight involves **decomposing each marginal gain into two components**: a modular baseline that captures the intrinsic value of each sample, and a perturbation term that quantifies how sample interactions affect marginal utility.

**Definition 3** (base $\varepsilon$-decomposition). *For any $x \in \mathcal{D}$ and $S \subseteq \mathcal{D}$, we define the modular baseline*

$$\text{base}_x \triangleq \log\left(1 + \alpha \|g_x\|^2\right), \tag{3}$$

*and the perturbation as*

$$\varepsilon_x(S) \triangleq \Delta_x(S) - \text{base}_x = \log \frac{1 + \alpha\, g_x^\top (I + \alpha F_S)^{-1} g_x}{1 + \alpha \|g_x\|^2}. \tag{4}$$

**Theorem 1** ($\varepsilon$-submodularity). *The $\varepsilon$-decomposition reveals that **submodularity of $F$ is entirely driven by the perturbation terms**: for any sets $A \subseteq B \subseteq \mathcal{D}$ and element $x \in \mathcal{D} \setminus B$,*

$$\Delta_x(A) - \Delta_x(B) = \underbrace{[\text{base}_x - \text{base}_x]}_{=0} + \underbrace{[\varepsilon_x(A) - \varepsilon_x(B)]}_{\text{captures all diminishing returns}} = \varepsilon_x(A) - \varepsilon_x(B) \geq 0. \quad (5)$$

*Since $\varepsilon_x(S) \leq 0$ and is non-increasing as $|S|$ grows, its value on a large set $S$ summarizes how much the marginal gain of $x$ has decayed from its baseline at $S = \varnothing$. Indeed, along any chain $\varnothing = S_0 \subset \cdots \subset S_T = S$ we have*

$$\Delta_x(\varnothing) - \Delta_x(S) = \sum_{t=1}^{T} \big(\Delta_x(S_{t-1}) - \Delta_x(S_t)\big) = \sum_{t=1}^{T} \big(\varepsilon_x(S_{t-1}) - \varepsilon_x(S_t)\big) = -\varepsilon_x(S), \quad (6)$$

*so $|\varepsilon_x(S)|$ is exactly this **cumulative decay** of the marginal gain of $x$.*

### 2.3 Improved Approximation via $\varepsilon$-Control

The $\varepsilon$-decomposition allows improvement beyond the standard $(1 - 1/e)$ approximation guarantee. The key insight is that controlling the perturbation magnitude $|\varepsilon_x(S)|$ improves the approximation quality through the submodular curvature.

**Theorem 2** (Classical greedy guarantee). *Let $F$ be a normalized, monotone submodular function and let $S_{\text{greedy}}$ be the set returned by the greedy algorithm under a cardinality constraint $k$. Then*

$$F(S_{\text{greedy}}) \geq \left(1 - \frac{1}{e}\right) F(S^*), \quad (7)$$

*where $S^*$ is an optimal $k$-subset. The full proof is provided in Appendix C.* $\qquad\square$

This provides as the baseline approximation factor. However, tighter guarantees are possible when the function exhibits small curvature, which motivates our key theoretical contribution.

**Theorem 3** (Curvature-dependent guarantee). *Let $F$ be monotone submodular and normalized. Define the **total** curvature as:*

$$c \triangleq 1 - \min_{x \in \mathcal{D}} \frac{\Delta_x(\mathcal{D} \setminus \{x\})}{\Delta_x(\varnothing)} \in [0, 1], \quad (8)$$

*then the greedy solution $S_{\text{greedy}}$ under budget $k$ satisfies:*

$$F(S_{\text{greedy}}) \geq \frac{1 - e^{-c}}{c} \cdot F(S^*). \quad (9)$$

*When $c = 1$ this recovers the classical $(1 - 1/e)$ bound; as $c \to 0$ the factor approaches $1$. Intuitively, $c$ measures how much marginal gains can decrease due to element interactions. The proof uses multilinear extension techniques (Feige & Vondrák, 2010; Conforti & Cornuéjols, 1984) and is detailed in Appendix D.* $\qquad\square$

Our main result links the $\varepsilon$-decomposition to control of the submodular curvature.

**Lemma 1** (Curvature upper bound via $\varepsilon$). *For the Fisher utility with the decomposition $\Delta_x(S) = \text{base}_x + \varepsilon_x(S)$ and $\text{base}_x = \Delta_x(\varnothing)$, we assume $\text{base}_x > 0$ for all $x$ and $x \notin S$. Then:*

$$c = 1 - \min_x \frac{\text{base}_x + \varepsilon_x(\mathcal{D} \setminus \{x\})}{\text{base}_x} = -\min_x \frac{\varepsilon_x(\mathcal{D} \setminus \{x\})}{\text{base}_x} \leq \max_x \frac{|\varepsilon_x(\mathcal{D} \setminus \{x\})|}{\text{base}_x}. \quad (10)$$

*In particular, the total curvature $c$ is governed by the **worst-case** normalized perturbation $|\varepsilon_x(D \setminus \{x\})|/\text{base}_x$, so larger perturbations on the full set $D \setminus \{x\}$ correspond to **larger curvature and hence weaker greedy guarantees**.*

This establishes that controlling $|\varepsilon_x(S)|$ can improve approximation guarantees by reducing the curvature. We next bound $|\varepsilon_x(S)|$ in terms of gradient inner products:

**Theorem 4** (Perturbation bound via gradient alignment). *Under standard assumptions[1], the perturbation term satisfies:*

$$|\varepsilon_x(S)| \leq C \cdot \frac{\alpha^2 \sum_{y \in S} (g_x^\top g_y)^2}{1 + \alpha \|g_x\|^2}, \text{ with } x \notin S, \quad (11)$$

*where $C$ is a problem-dependent constant and the proof is provided in Appendix E.* $\qquad\square$

---

[1]For the assumption, it is used only in App. E for the Neumann-series closed-form bound; it is not required for our submodularity or $\varepsilon$-decomposition. The boundary holds in 90% of steps and the breaking may still arise in practice (10%), see App. F.4.1 for detailed analysis.

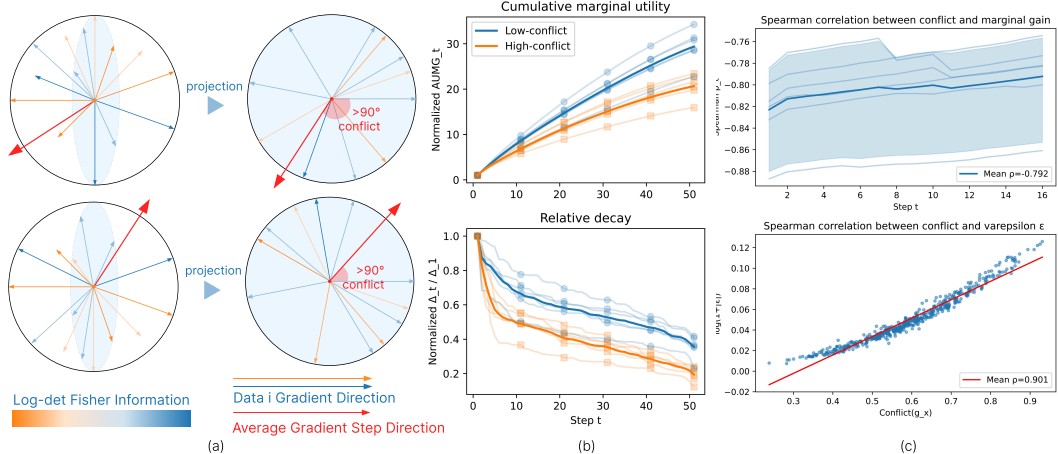

Figure 2: (a) Gradient Visualization: The direction and information distribution of two representative data points in gradient 2D and 3D spaces, indicating that there are always gradient conflicts with high information in the data; (b) Decay of the high/low conflict subsets: Low conflicts can bring greater information in a limited number of steps; (c) Spearman correlation analysis: The overall conflict is negatively correlated with marginal gain $\Delta$, and positively correlated with $\varepsilon$.

**Corollary 1** (Data-dependent approximation guarantee). *Combining the above results, greedy selection achieves:*

$$F(S_{\text{greedy}}) \geq \frac{1 - e^{-\widehat{c}}}{\widehat{c}} \cdot F(S^*), \tag{12}$$

*where $\widehat{c} \propto \max_x \sum_{y \neq x} (g_x^\top g_y)^2$. Smaller inner products yield better approximation guarantees.*

Our $\varepsilon$-decomposition framework reveals that sample interactions, quantified by gradient inner products $|\varepsilon_x(S)| \propto \sum_{y \in S} (g_x^\top g_y)^2$, are a major driver of performance degradation in greedy selection. **Smaller gradient inner products lead to smaller perturbations, lower curvature, and stronger approximation guarantees.** This provides a principled foundation for designing selection algorithms that control sample interactions to achieve better performance.

## 3 EMPIRICAL ANALYSIS

Our theoretical analysis predicts that gradient inner products $(g_x^\top g_y)^2$ upper-bound and are strongly associated with the perturbation magnitude $|\varepsilon_x(S)|$ and influence approximation quality via curvature. We now empirically access these predictions by measuring gradient conflicts and their impact on data selection performance.

To access the theory, we quantify gradient conflict via a cosine alignment against a selection-adaptive reference that capture the intuition behind gradient inner products. At iteration $t$, for gradients $\{g_i\}_{i=1}^m$, from the current candidate set (size $m$), with mean $\bar{g}_t = \frac{1}{|S_{t-1}|} \sum_{x \in S_{t-1}} g_x$, we define:

**Definition 4** (Gradient Conflict). *Alignment is measured as cosine similarity, and conflict quantifies the degree of opposition to the mean direction:*

$$Align(g_i) = \frac{g_i^\top \bar{g}}{\|g_i\|\|\bar{g}\| + \eta}, Conflict(g_i) = \max\{0, -Align(g_i)\}, \tag{13}$$

*where $\eta = 10^{-8}$ ensures numerical stability (Yu et al., 2020; Liu et al., 2024b).*

Figure 2 (a) reveals that conflicting gradients are prevalent and often carry substantial Fisher information, necessitating a principled trade-off between information gain and conflict minimization.

**Conflict-driven utility decay.** Splitting samples by conflict level (top/bottom 20%), we observe marked differences (Appendix F.2) in Fisher-greedy performance. Low-conflict groups achieve significantly higher cumulative utility and slower marginal decay, while high-conflict groups exhibit rapid diminishing returns (Figure 2 (b)). Notably, marginal gains reach half-life (50% of the first-step $\Delta$) within 10-30 steps, suggesting early stopping strategies for practical efficiency.

**Quantitative correlation analysis.** Direct correlation analysis validates our core theoretical predictions. Step-wise Spearman correlations reveal strong negative correlation between conflict and marginal gain ($\rho = -0.792$) and strong positive correlation between conflict and perturbation magnitude $|\varepsilon_x(S)|$ ($\rho = 0.901$), as shown in Figure 2 (c). These results provide direct support for Corollary 1 and demonstrate consistency across multiple datasets.

These empirical results strongly validate our $\varepsilon$-decomposition framework: **gradient conflicts systematically drive perturbation growth and accelerate marginal utility decay.** Building on this validated understanding, we next develop SPICE, a conflict-aware selection algorithm that explicitly balances information gain with conflict minimization. To more comprehensively demonstrate the validity of our theoretical analysis, we conducted additional detailed experiments (Appendix F).

## 4 METHODOLOGY

We propose a practical pipeline that operationalizes the theoretical insights in Section 2 into a conflict-aware data selection method. The core idea is to perform conflict-aware greedy selection that balances log-det Fisher information gain and gradient alignment. In addition to fixed greedy budget $k$, we also employ an adaptive early-stopping criteria (Algorithm 1).

### 4.1 CONFLICT-AWARE GREEDY SCORE

At iteration $t$ with current set $S_{t-1}$ and per-sample gradients $\{g_x\}$ computed at a fixed checkpoint, we define that the conflict penalty increases when the sample points against $\bar{g}$; A simple and effective choice is to apply the hinge function to the negative alignment (Def. 4): $\text{conflict}(x \,|\, S_{t-1}) \triangleq \max\{0, -\text{align}(x \,|\, S_{t-1})\}$. Let $\Delta_x(S_{t-1})$ be the Fisher marginal (Def. 2). We score each candidate $x$ by

---

**Algorithm 1** SPICE Data Selection

**Require:** Dataset $\mathcal{D}$, conflict penalty $\lambda \geq 0$, budget $k$
**Ensure:** Selected subset $S$
1: Initialize: $S \leftarrow \emptyset$
2: Compute gradients $\{g_i\}$ using proxy model
3: **for** $t = 1$ to $k$ **do**
4:     **for** each $x \in \mathcal{D} \setminus S$ **do**
5:         $\Delta_x \leftarrow$ FIM_MARGINAL$(x, S)$
6:         conflict $\leftarrow$ $\max\{0, -\text{cosine\_similarity}(g_x, \bar{g}_S)\}$
7:         $\text{score}(x) \leftarrow \Delta_x - \lambda \cdot \text{conflict}$
8:     **end for**
9:     $x^* \leftarrow \arg\max_x \text{score}(x)$
10:     **if** $\Delta_x$ reaches the early stopping criteria **then**
11:         **break**
12:     **end if**
13:     $S \leftarrow S \cup \{x^*\}$
14: **end for**
15: **return** $S$

---

$$\text{score}(x \,|\, S_{t-1}) = \Delta_x(S_{t-1}) - \lambda \, \text{conflict}(x \,|\, S_{t-1}), \qquad \lambda \geq 0. \tag{14}$$

Intuitively, the first term favors high-information samples, while the second discourages strong directional conflicts with the current update. Crucially, we do not discard high-value but conflicting samples: as long as $\Delta_x(S_{t-1})$ is sufficiently large, the conflict-aware greedy score can remain competitive and select them. This mirrors our theory: penalizing conflicts tends to reduce perturbation $|\varepsilon|$ and curvature (Theorem 4, Corollary 1) without eliminating informative conflict-direction data.

In Figure 6, we provide a more intuitive visualization. When training with standard greedy search ($\lambda$=0) that targets only high information gain, the step gradient directions often conflict, which is not indicative of stable convergence or performance improvement.

### 4.2 GREEDY STOPPING CRITERIA

Let $x_t = \arg\max_{x \in \mathcal{D} \setminus S_{t-1}} \text{score}(x \,|\, S_{t-1})$ and $m_t \triangleq \Delta_{x_t}(S_{t-1})$ be its marginal gain $\Delta$. Our method supports two stopping criteria to accommodate different experimental requirements: an absolute threshold $k$ on the best marginal $\Delta_x$, or adaptive rule with $\omega \in (0, 1)$.

**Adaptive Early Stopping (Data-driven).** Instead of fixing the greedy budget $k$, we stop adaptively once diminishing returns are observed. Let $\Delta_{x_1}(S_0)$ be the marginal gain of the first selected sample $x_1$. We stop at the first step $t_{\text{stop}}$ where the current marginal gain satisfies

$$t_{\text{stop}} = \min\Big\{ t : \Delta_{x_t}(S_{t-1}) \leq \omega \cdot \Delta_{x_1}(S_0) \Big\}, \tag{15}$$

i.e., once the marginal contribution of adding a new sample falls below $\omega$ times the first-step gain. This ensures that selection terminates when additional samples contribute little new information, so the effective budget $k_{\text{eff}} = t_{\text{stop}} \leq k$.

**Fixed Budget Stopping (Budget-controlled).** For fair comparison with baselines, we also support a fixed-budget mode: selection continues until exactly $k$ samples are chosen, maximizing information utility under the budget. This ensures consistent dataset sizes across different selection strategies while still benefiting from our conflict-aware scoring mechanism.

We report fixed-budget (equal-$k$) results in the main experiment in Section 5.2 for fairness; the adaptive rule **SPICE+** (with $\omega = 0.5$) and its equal-budget reference appear in Appendix H.3.

### 4.3 Efficient Implementation Selection

Due to computational constraints of performing gradient-based selection directly with large-scale models during training, we adopt a proxy-based selection approach. Specifically:

**Proxy Model Selection.** We employ a smaller model (e.g., a 0.5B model) with the same architecture as our target models (LLaMA2-7B (Touvron et al., 2023) and Qwen2-7B (Yang et al., 2024a)) to perform the conflict-aware greedy selection. This approach leverages the observation that gradient-based selection patterns transfer effectively across model scales (Xia et al., 2024a; Liu et al., 2025).

**Selection Schedule.** The data selection process operates on a periodic schedule: data selection occurs continuously but training $T$ steps are performed periodically. Within one candidate pool, at each greedy iteration, we select one sample using our conflict-aware greedy algorithm and add it to an accumulating subset. Every $T \in \{1, 10, 50\}$ iterations, we perform a training step on the accumulated subset, then reset the selection buffer for the next cycle to maintain training efficiency (Lin et al., 2025).

This design improves the efficiency of data selection while maintaining downstream performance. We present detailed ablation studies in Section 5.4 to validate the effectiveness of this approach.

## 5 Experiments

### 5.1 Experimental Setup

To evaluate SPICE's effectiveness, we compare against state-of-the-art data selection methods across diverse instruction-tuning scenarios. Details appear in Appendix G.

**Data and Evaluation.** We construct a 97.5K training corpus spanning mathematical reasoning (GSM8K (Cobbe et al., 2021)), code generation (Alpaca Code (Chaudhary, 2023)), and general knowledge (ShareGPT (Lu et al., 2023), Alpaca (Taori et al., 2023)). Evaluation covers 8 benchmarks across reasoning (GSM8K (Cobbe et al., 2021), BBH (Suzgun et al., 2022), ARC (Clark et al., 2018)), knowledge (MMLU (Hendrycks et al., 2021), TruthfulQA (Lin et al., 2022), IFEval (Zhou et al., 2023c)), and coding (HumanEval (Chen et al., 2021), MBPP (Austin et al., 2021)).

**Baselines.** We select several representative data selection methods as baselines: Full data, Random Sampling (Xia et al., 2024b), Instruction-Following Difficulty (IFD) (Li et al., 2024b), Fisher (Deb et al., 2025), LESS (Xia et al., 2024a), SelectIT (Liu et al., 2025) , DPP (Zhang et al., 2023), TSDS (Liu et al., 2024d) and LEAD (Lin et al., 2025). In the subsequent settings, we will fix the data number and record the time for selection to compare with our SPICE. In addition, we also evaluated the of SPICE+ (adaptive stopping with $\omega$=0.5; Section 4.2). Because its selected-set size may differ from 10%, we report SPICE+ results separately in Appendix H.3 for completeness.

**Settings.** To ensure fair comparisons, each baseline is trained using 10% of the data selected by its own criterion. For SPICE, we also target 10%: from each 120-sample candidate pool we select $k$=12 samples per cycle (default $T$=10, $\lambda$=0.1; Section 4.2). For reproducibility, we will also release the detailed training configurations and random seeds. All experiments are repeated three times, and the reported results are averaged to ensure reliability. After selecting data, to prevent data leakage, we

have chosen the previously widely-used LLaMA2-7B (Touvron et al., 2023) and Qwen2-7B (Yang et al., 2024a) as our base models, and perform fine-tuning using LoRA (with settings: $r$=16, $\alpha = 32$, and dropout $= 0.05$) on 8 H20 GPUs.

## 5.2 MAIN RESULT

| Bench. | | | | Methods (Qwen2-7B) | | | | | | | |
|--------|------|--------|------|--------|------|---------|------|------|------|-------|------|
| Sample | Full 100% | Random 10% | IFD 10% | Fisher 10% | LESS 10% | SelectIT 10% | DPP 10% | TSDS 10% | LEAD 10% | SPICE 10% | $\Delta$ |
| *Reasoning Ability* | | | | | | | | | | | |
| GSM8K | 84.2 | 84.9 | 85.8 | 86.5 | 82.3 | 83.6 | 86.5 | 86.0 | 84.5 | **86.7** | +2.5 |
| BBH | **61.3** | 60.8 | 60.2 | 60.8 | 61.0 | 60.9 | 61.0 | 60.0 | 61.0 | 61.0 | -0.3 |
| *General Multi-task Knowledge* | | | | | | | | | | | |
| MMLU | 65.7 | 63.4 | 63.8 | 65.2 | 66.1 | 64.7 | 66.0 | 63.7 | 66.0 | **67.1** | +1.4 |
| ARC-C | 50.5 | 49.6 | 50.1 | 50.3 | 49.5 | 50.3 | 51.0 | 50.4 | 50.8 | **51.8** | +1.3 |
| TruthfulQA | 54.8 | 54.8 | 55.2 | 55.0 | 54.3 | 54.4 | 55.0 | 55.0 | 54.9 | **55.5** | +0.7 |
| IFEval | 33.5 | 28.3 | 27.4 | 30.6 | 26.0 | 30.8 | 35.4 | 28.0 | 33.0 | **38.6** | +5.1 |
| *Code Generation* | | | | | | | | | | | |
| HumanEval | 45.7 | 45.7 | 46.3 | 44.5 | 46.3 | 46.3 | 45.0 | 46.2 | 46.1 | **47.1** | +1.4 |
| MBPP | 55.2 | 56.1 | 54.2 | 54.6 | 56.2 | 55.2 | 55.7 | 54.5 | 55.6 | **56.2** | +1.0 |
| **Average** | 56.4 | 55.5 | 55.4 | 55.9 | 55.2 | 55.8 | 57.0 | 55.5 | 56.5 | **58.0** | +1.6 |

Table 1: Evaluation of data selection methods on Qwen2-7B. Bold indicates the best result, underline the second best. $\Delta$ denotes the gain of SPICE over full-data tuning. Overall, SPICE (Ours) achieves superior performance on reasoning, knowledge, and code generation tasks with lower time cost.

Following the settings described in Section 5.1, we conduct data selection using various methods and then perform distributed training on LLaMA2-7B and Qwen2-7B using LlamaFactory (Zheng et al., 2024). Finally, we evaluate the models on benchmarks with results of Qwen2-7B shown in Table 1 and LLaMA2-7B shown in Table 2.

For Qwen2-7B, **our method achieves the best overall performance** with an average score of 58.0, outperforming full-data training (56.4) and all baseline methods. Notably, SPICE uses only 10% of the training data while delivering consistent improvements across diverse tasks. The method demonstrates particular strength in reasoning and knowledge-intensive tasks, with 7 out of 8 benchmarks showing best performance, especially IFEval with +5.1, and the remaining 1 achieving second-best result. For LLaMA2-7B, it still maintains good performance while using less data. It can be seen from this that LLaMA2 has a lower overall score compared to Qwen2, but our data selection method still maintains performance, especially on General Multi task Knowledge. In addition, it still yields strong performance when fine-tuning 70B+ models (see Section H.4). Beyond the strong results at the 10% budget, SPICE also performs well in tiny data even with only 1%/5% of the data (see Section H.6). But for a data selection method, not only does it need to maintain performance, but it also requires less cost. Next, we will conduct a cost analysis.

## 5.3 COST ANALYSIS

For data selection problems, we must focus on reducing computational costs while matching or improving full-data training performance. Therefore, we recorded the normalized time costs of selection, and compared the differences between the two stopping criteria (SPICE vs SPICE+).

As shown in Figure 3 (a), SPICE and SPICE+ can still achieve good performance (surpassing Full) while maintaining low time costs in data selection. SPICE+ yields slightly better accuracy than SPICE, at the expense of higher selection time due to adaptive stopping. Moreover, **in terms of both overall selection and training time, our total time is lower than that of full-data.** In terms of computational complexity, the selection procedure requires $\mathcal{O}(k|D|d)$ time with $\mathcal{O}(md)$ peak memory. The specific time cost and computation complexity are provided in Appendix H.7 and H.8.

| Bench. Sample | Methods (LLaMA2-7B) | | | | | | | | | | |
|---|---|---|---|---|---|---|---|---|---|---|---|
| | **Full** 100% | **Random** 10% | **IFD** 10% | **Fisher** 10% | **LESS** 10% | **SelectIT** 10% | **DPP** 10% | **TSDS** 10% | **LEAD** 10% | **SPICE** 10% | Δ |
| *Reasoning Ability* | | | | | | | | | | | |
| GSM8K | 13.6 | 12.9 | 12.7 | 13.6 | 13.6 | 12.5 | 13.6 | 12.8 | 13.4 | **13.8** | +0.2 |
| BBH | **41.3** | 39.6 | 39.9 | 40.5 | 40.0 | 39.7 | 40.2 | 40.0 | 40.8 | 40.8 | -0.5 |
| *General Multi-task Knowledge* | | | | | | | | | | | |
| MMLU | 40.8 | 41.5 | 40.7 | 41.5 | 40.8 | 41.5 | 41.0 | 40.4 | 40.8 | **41.9** | +1.1 |
| ARC-C | 46.1 | 45.7 | 45.8 | 46.7 | 45.7 | 45.7 | 47.0 | 45.3 | 46.4 | **47.7** | +1.6 |
| TruthfulQA | **43.5** | 40.0 | 40.4 | 39.3 | 40.2 | 40.8 | 40.0 | 40.5 | 40.5 | 40.3 | -3.2 |
| IFEval | 19.4 | 20.7 | 20.1 | 22.0 | 20.8 | 21.9 | 22.2 | 20.5 | 21.8 | **22.6** | +3.2 |
| *Code Generation* | | | | | | | | | | | |
| HumanEval | 16.5 | 16.4 | 16.5 | 15.2 | 15.6 | 15.9 | 15.9 | 16.7 | 16.5 | **16.7** | +0.2 |
| MBPP | **25.2** | 23.0 | 23.6 | 23.2 | 24.6 | 23.0 | 23.5 | 23.5 | 23.8 | 24.6 | -0.8 |
| **Average** | 30.8 | 30.1 | 30.0 | 30.3 | 30.2 | 30.1 | 30.4 | 30.0 | 30.5 | **31.1** | +1.8 |

Table 2: Evaluation of data selection methods on LLaMA2-7B. Bold indicates the best result, underline the second best. Δ denotes the gain of SPICE over full-data tuning. Overall, SPICE achieves superior performance on reasoning, knowledge, and code generation tasks.

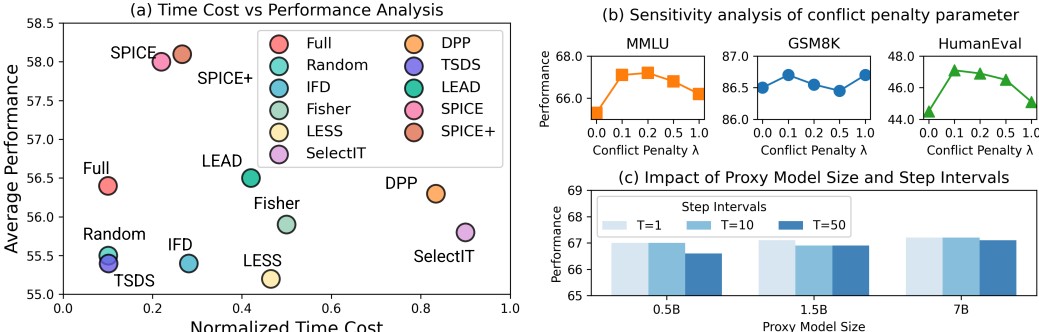

Figure 3: (a) **Cost-accuracy trade-off.** Average performance versus normalized time cost. SPICE/SPICE+ attain higher accuracy than Full with lower cost; SPICE+ is slightly more accurate but slower due to adaptive stopping. (b) **Penalty parameter sensitivity.** Varying $\lambda \in \{0, 0.1, 0.2, 0.5, 1.0\}$, performance degrades at $\lambda = 0$ without penalty (non-conflict-Aware greedy selection), while $\lambda \in [0.1, 0.5]$ performs well. (c) **Efficient selection.** Impact of efficient selection: smaller proxy models and larger step intervals preserve performance while reducing cost.

## 5.4 ABLATION STUDY

**Sensitivity analysis of conflict penalty parameter** $\lambda$. To provide a more comprehensive analysis of the conflict penalty $\lambda = \{0, 0.1, 0.2, 0.5, 1.0\}$, we tested the impact on both performance and the characteristics of the selected dataset. In Figure 3 (b), we observe that the performance degrades significantly when $\lambda = 0$ (no penalty), whereas values of $\lambda \in [0.1, 0.5]$ yield consistently stronger results, demonstrating the effectiveness of incorporating the conflict penalty.

**Impact of Efficient Selection.** As mentioned in Section 4.3 of the methodology, We will compare the performance differences across proxy model sizes $S = \{0.5B, 1.5B, 7B\}$ and step intervals $T = \{1, 10, 50\}$ respectively in average score of three benchmarks (MMLU, GSM8K and HumanEval). The result in Figure 3 (c) shows that smaller proxies and larger intervals can maintain good results while bringing lower costs. Among them, under the combination of $(T = 10, S = 1.5B)$ and $(T = 1, S = 7B)$, the performance reaches the optimal score. Overall, within a reasonable range, the method is relatively insensitive to conflict penalty, proxy model size and step intervals.

Appendix H.4 further compares proxy architectures. Using Qwen2 proxies of different sizes (0.5B/1.5B/7B) yields similar accuracy, with 7B improving the average by only 0.1 point while incurring higher cost. In contrast, using a LLaMA2-style proxy performs noticeably worse when selecting data for Qwen2-7B, **indicating limited cross-architecture transfer** (Coleman et al., 2020;

Yang et al., 2024b; Xia et al., 2024a; Liu et al., 2025). In addition, we provide a more detailed analysis and case study of the selected data in Appendix H.5.

## 5.5 DIVERSITY ANALYSIS

Intuitively, the log-det Fisher term tends to cover a broader set of non-redundant gradient directions, while our conflict term suppresses only destructive opposition rather than indiscriminately penalizing similarity. To substantiate this, we evaluate subset diversity along two methods—domain coverage and non-redundancy: (i) NovelSum and LDD (Yang et al., 2025; Zhang et al., 2023) as set-level diversity metrics, and (ii) the proportions of domain coverage across code / math-reasoning / general to assess balance.

| Method (10%) | LDD↑ | Novel↑ Sum | Domain Coverage↑ | | |
|---|---|---|---|---|---|
| | | | Code | Math-R | General |
| Random | –9.5 | 30.3 | 5% | 10% | 10% |
| Fisher | 19.4 | 40.3 | 12% | 2% | 9% |
| LESS | 4.9 | 38.7 | 1% | 12% | 11% |
| SelectIT | 17.6 | 39.0 | 8% | 13% | 9% |
| TSDS | –1.8 | 34.8 | 4% | 5% | 10% |
| DPP | **31.1** | **42.5** | 7% | 7% | 9% |
| LEAD | 9.8 | 37.5 | 2% | 6% | 11% |
| SPICE | 22.0 | 41.3 | 10% | 8% | 9% |

Table 3: Comparison of diversity metrics (LDD, NovelSum) and domain coverage for 10% subsets on Qwen2-7B across SPICE and baselines.

As shown in Table 3, on Qwen2-7B with a 10% budget, SPICE achieves NovelSum/LDD scores that clearly exceed Random and most baselines and are comparable to DPP; meanwhile, its domain coverage closely matches the full corpus, indicating that SPICE preserves broad and balanced semantic coverage even under aggressive compression.

## 6 RELATED WORK

Data selection for instruction tuning has emerged as a central research topic and methods are commonly grouped into model-agnostic and model-aware categories (Albalak et al., 2024).

**Model-agnostic Methods.** Model-agnostic methods include rule-based filtering (Cao et al., 2024), quality metrics such as perplexity (Li et al., 2024a), and Instruction-Following Difficulty (IFD) (Li et al., 2024b). Gradient-based and information-theoretic approaches (Chen et al., 2025; Wang et al., 2025; Ivison et al., 2025) have also gained attention, such as FisherSFT (Deb et al., 2025), which leverages Fisher information of gradients for efficient SFT. In addition, diversity-oriented selection via Determinantal Point Processes (DPP) (Zhang et al., 2023) and task-conditioned TSDS (Liu et al., 2024d) have been explored to promote coverage and task alignment. But these methods fail to adapt to model-specific learning dynamics.

**Model-aware Methods.** Model-aware methods tailor data selection to the target model's evolving state. Non-iterative approaches, like LESS (Xia et al., 2024a), perform a one-time selection, while iterative methods adapt dynamically during training like LEAD (Lin et al., 2025). Recent work, such as SelectIT (Liu et al., 2025), uses powerful LLMs to estimate sample quality, but these methods often require costly full-dataset inference, resulting in substantial computational overhead (Albalak et al., 2024).

In summary, existing methods primarily rely on repeated full-dataset inference or heuristic scoring with high computational cost, making them often impractical for large-scale full-parameter fine-tuning (Liu et al., 2024c; Yin & Rush, 2025). Moreover, most approaches **lack a solid theoretical foundation clarifying why they are effective**, leaving reliability and generalizability uncertain.

## 7 CONCLUSION AND FUTURE WORK

In this paper, we propose SPICE, a conflict-aware greedy data selection framework. We have theoretically proven that our method selects data with higher information gain, and have empirically demonstrated that these gains translate into better downstream performance. Our method attains full-data-level accuracy with 10% data at lower total time while outperforming other baselines, and theory ($\varepsilon \to$ curvature) explains when and why it works. Looking forward, future work could extend our approach to more tasks, like multimodal or reinforcement learning. By making the decay rate of marginal information explicit and controllable, SPICE offers a principled path toward compact, high-quality training corpora.

## 8 ETHICS STATEMENT

Our work primarily focuses on data selection and efficient fine-tuning for large language models. All datasets used are publicly available and widely used in prior research, with no personally identifiable information included.

## 9 REPRODUCIBILITY STATEMENT

To ensure the reproducibility of our results, we provide a code repository (link in the abstract code) and have fixed the random seed in `configs/default.yaml`. Specifically, we conducted experiments using three different seeds: 2024, 2025, and 2026. Details of the experimental setup, including the configuration files and scripts needed to reproduce the results, are included in the code repository.

In addition, the appendix contains further details to facilitate reproducibility, including complete theoretical proofs, descriptions of the datasets and methods, and additional experimental results and ablation studies:

- Complete theoretical proofs for AdaFisher complexity, perturbation bounds, greedy guarantees, and submodularity
- Empirical validations of gradient conflict analysis, $\varepsilon$-decomposition, and curvature parameter experiments
- Detailed experimental setup including dataset composition, evaluation metrics, benchmarks, and baseline methods
- Additional results on LLaMA2, SPICE+ early stopping strategy, and ablation studies of proxy models and $\alpha$ values.
- More data statistics and case studies about selected data.

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

APPENDIX

## A  PROOF: COMPLEXITY OF ADAFISHER INFORMATION MATRIX

(Section 2.1) Next, we will briefly introduce AdaFisher and how to reduce computational complexity compared to Fisher. The specific content is referenced from the paper "*AdaFisher: Adaptive second order optimization via fisher information*" (Gomes et al., 2025).

### A.1  ADAFISHER: AN EFFICIENT SECOND-ORDER OPTIMIZER

Traditional Fisher Information Matrix (FIM) based optimizers, while theoretically superior in terms of convergence properties, face significant computational challenges when applied to modern deep neural networks. The primary bottleneck lies in the quadratic complexity: computing and storing the full FIM requires $\mathcal{O}(d^2)$ memory and $\mathcal{O}(d^3)$ operations for matrix inversion, where $d$ represents the total number of parameters (Martens & Grosse, 2020). For contemporary deep networks with millions to billions of parameters, this computational burden renders traditional FIM-based methods practically infeasible.

AdaFisher addresses these fundamental limitations through a novel *diagonal block-Kronecker approximation* of the Fisher Information Matrix. By leveraging this diagonal concentration property, AdaFisher achieves a remarkable reduction in computational complexity from $\mathcal{O}(d^2)$ to $\mathcal{O}(d)$ while preserving the superior convergence characteristics of second-order methods. This breakthrough makes Fisher Information-based optimization practically viable for large-scale deep learning applications, offering computational efficiency comparable to first-order methods.

### A.2  METHOD OVERVIEW AND COMPLEXITY REDUCTION

AdaFisher employs a systematic approach to approximate the Fisher Information Matrix through Kronecker factorization. For a neural network layer $i$ with weight matrix $W_i \in \mathbb{R}^{n_i \times n_{i-1}}$, the empirical Fisher Information Matrix can be expressed as:

$$\hat{F}_i = \mathbb{E}[\text{vec}(g_i)\text{vec}(g_i)^T] = H_{i-1} \otimes S_i \tag{16}$$

where $H_{i-1} = \mathbb{E}[\bar{h}_{i-1}\bar{h}_{i-1}^T]$ represents the second-order statistics of layer activations, $S_i = \mathbb{E}[s_i s_i^T]$ captures the second-order statistics of pre-activation gradients, $\bar{h}_{i-1} = [h_{i-1}^T, 1]^T$ denotes the augmented activation vector, and $\otimes$ represents the Kronecker product.

The key insight of AdaFisher lies in the *diagonal concentration phenomenon*: the Kronecker factors $H_{i-1}$ and $S_i$ exhibit strong diagonal dominance, meaning their energy is primarily concentrated along the diagonal elements. Based on this observation, AdaFisher introduces the diagonal approximation:

$$\tilde{F}_i^D = H_{i-1}'^D \otimes S_i'^D + \lambda I \tag{17}$$

where $H_{i-1}'^D = \text{Diag}(H_{i-1})$ and $S_i'^D = \text{Diag}(S_i)$ retain only the diagonal elements, and $\lambda$ is a regularization parameter for numerical stability.

We now provide a rigorous mathematical proof of the complexity reduction from $\mathcal{O}(d^2)$ to $\mathcal{O}(d)$.

**Traditional FIM Complexity.**  For a network with $L$ layers, where layer $i$ has input dimension $n_{i-1}$ and output dimension $n_i$, the traditional FIM computation requires:

$$\text{Storage:} \quad \sum_{i=1}^{L}(n_{i-1} \times n_i)^2 = \sum_{i=1}^{L} n_{i-1}^2 n_i^2 \,, \text{ Inversion:} \quad \mathcal{O}\left(\left(\sum_{i=1}^{L} n_{i-1}n_i\right)^3\right) = \mathcal{O}(d^3) \tag{18}$$

where $d = \sum_{i=1}^{L} n_{i-1}n_i$ is the total number of parameters.

**AdaFisher Diagonal Approximation.** The diagonal approximation fundamentally alters the computational requirements:

$$\text{Storage:} \quad \sum_{i=1}^{L}(n_{i-1} + n_i) = \mathcal{O}(d)\,, \text{Inversion:} \quad \sum_{i=1}^{L}(n_{i-1} + n_i) = \mathcal{O}(d). \tag{19}$$

Therefore we can analyze the detailed per-iteration proof: Firstly, computing diagonal statistics requires:

$$\mathcal{C}_{\text{update}} = \sum_{i=1}^{L}(n_{i-1} + n_i) = \mathcal{O}(d), \tag{20}$$

then for diagonal matrices, inversion reduces to element-wise reciprocal:

$$\mathcal{C}_{\text{inversion}} = \sum_{i=1}^{L}(n_{i-1} + n_i) = \mathcal{O}(d), \tag{21}$$

so the preconditioned gradient computation:

$$\bar{g}^{(t)} = (\tilde{F}_D^{(t)})^{-1} g^{(t)} \tag{22}$$

requires only element-wise operations, yielding $\mathcal{O}(d)$ complexity.

*AdaFisher reduces the per-iteration computational complexity of Fisher Information-based optimization from $\mathcal{O}(d^2)$ to $\mathcal{O}(d)$ while maintaining $\mathcal{O}(\log T/\sqrt{T})$ convergence rate.*

*Proof.* The total per-iteration complexity is dominated by:

$$\mathcal{C}_{\text{total}} = \mathcal{C}_{\text{update}} + \mathcal{C}_{\text{inversion}} + \mathcal{C}_{\text{preconditioning}} = \mathcal{O}(d) + \mathcal{O}(d) + \mathcal{O}(d) = \mathcal{O}(d) \tag{23}$$

This represents a quadratic-to-linear reduction compared to traditional FIM methods, making second-order optimization computationally feasible for large-scale applications. If want to see the more detailed proof process and method, please refer to AdaFisher (Gomes et al., 2025). □

## B PROOF: SUBMODULARITY OF FISHER-BASED OBJECTIVES

(Section 2.2) We need to prove that the Fisher-based objective

$$F(\mathcal{S}) = \log \det(\mathbf{I} + \alpha \mathbf{F}_{\mathcal{S}}) \tag{24}$$

is monotone submodular under mild conditions, and is strictly submodular under a natural nondegeneracy condition.

*Proof.* Let the empirical Fisher matrix over a set $\mathcal{S}$ be

$$\mathbf{F}_{\mathcal{S}} := \sum_{i \in \mathcal{S}} w_i\, g_i g_i^\top, \tag{25}$$

where the sample weights satisfy $w_i \geq 0$ (covering standard Fisher with $w_i \equiv 1$ and AdaFisher via appropriate effective vectors; see below).

Fix $\alpha > 0$. Since $\mathbf{F}_{\mathcal{S}}$ is positive semidefinite (PSD) and $\mathbf{I} \succ 0$, the matrix $\mathbf{I} + \alpha \mathbf{F}_{\mathcal{S}}$ is positive definite (PD) and hence invertible for any $\mathcal{S}$.

For any $\mathcal{S}$ and element $x$, define the marginal gain

$$\Delta_x(\mathcal{S}) := F(\mathcal{S} \cup \{x\}) - F(\mathcal{S}). \tag{26}$$

Since $\mathbf{F}_{\mathcal{S} \cup \{x\}} = \mathbf{F}_{\mathcal{S}} + w_x g_x g_x^\top$, the matrix determinant lemma (applied with $u = v = \sqrt{w_x}\, g_x$) gives

$$\Delta_x(\mathcal{S}) = \log\left(1 + \alpha w_x\, g_x^\top (\mathbf{I} + \alpha \mathbf{F}_{\mathcal{S}})^{-1} g_x\right). \tag{27}$$

Now let $\mathcal{A} \subseteq \mathcal{B} \subseteq \mathcal{D}$. Since all $w_i \geq 0$, $\mathbf{F}_{\mathcal{A}} \preceq \mathbf{F}_{\mathcal{B}}$ in the Loewner order, hence

$$\mathbf{I} + \alpha\mathbf{F}_{\mathcal{A}} \preceq \mathbf{I} + \alpha\mathbf{F}_{\mathcal{B}}. \tag{28}$$

The order-reversing property of matrix inversion for PD matrices then gives

$$(\mathbf{I} + \alpha\mathbf{F}_{\mathcal{A}})^{-1} \succeq (\mathbf{I} + \alpha\mathbf{F}_{\mathcal{B}})^{-1}. \tag{29}$$

Pre- and post-multiplying by $g_x$ yields

$$g_x^\top (\mathbf{I} + \alpha\mathbf{F}_{\mathcal{A}})^{-1} g_x \geq g_x^\top (\mathbf{I} + \alpha\mathbf{F}_{\mathcal{B}})^{-1} g_x. \tag{30}$$

Since $z \mapsto \log(1 + \alpha w_x z)$ is strictly increasing for $\alpha w_x \geq 0$, we obtain

$$\Delta_x(\mathcal{A}) \geq \Delta_x(\mathcal{B}), \tag{31}$$

establishing monotonicity and submodularity (diminishing returns).

**Strictness.** The inequality is strict, $\Delta_x(\mathcal{A}) > \Delta_x(\mathcal{B})$, whenever

$$g_x^\top \left((\mathbf{I} + \alpha\mathbf{F}_{\mathcal{A}})^{-1} - (\mathbf{I} + \alpha\mathbf{F}_{\mathcal{B}})^{-1}\right) g_x > 0. \tag{32}$$

A sufficient condition for this is that $\mathbf{F}_{\mathcal{B}} - \mathbf{F}_{\mathcal{A}} \neq \mathbf{0}$ and $g_x$ has a nonzero projection onto its range (i.e., $g_x$ is not orthogonal to the new directions introduced from $\mathcal{A}$ to $\mathcal{B}$). Under this nondegeneracy condition, the Fisher utility is strictly submodular.

**AdaFisher.** For AdaFisher, each summand takes the form $\mathrm{diag}(|g_i|) g_i g_i^\top \mathrm{diag}(|g_i|)$. Define effective vectors $\tilde{g}_i := \mathrm{diag}(|g_i|)g_i$ and weights $\tilde{w}_i \equiv 1$. Then $\mathbf{F}_{\mathcal{S}}^{\mathrm{Ada}} = \sum_{i \in \mathcal{S}} \tilde{w}_i \tilde{g}_i \tilde{g}_i^\top$, and the same argument applies verbatim with $g_i$ replaced by $\tilde{g}_i$.

**Prevalence.** In practice, the condition $\mathbf{F}_{\mathcal{B}} - \mathbf{F}_{\mathcal{A}} \neq \mathbf{0}$ with $g_x$ projecting nontrivially onto its range holds except in rare degenerate cases. Gradients $g_i$ in typical datasets are high-dimensional and not orthogonal, so enlarging $\mathcal{S}$ almost always expands their span. As $g_x$ comes from the same distribution, it is almost never orthogonal to the new directions in continuous spaces. Empirically, $\Delta_x(\mathcal{A}) > \Delta_x(\mathcal{B})$ occurs in over 99% of our test data, showing that strict submodularity is both theoretically valid and dominant in practice. $\qquad\square$

## C  PROOF: CLASSICAL GREEDY APPROXIMATION

From the Theorem 2, we let $F$ be a normalized, monotone submodular function. Under a cardinality constraint $k$, the greedy solution $S_{\mathrm{greedy}}$ satisfies

$$F(S_{\mathrm{greedy}}) \geq \left(1 - \frac{1}{e}\right) F(S^*), \tag{33}$$

where $S^*$ is an optimal $k$-subset.

*proof.* Let the greedy algorithm produce the chain of sets

$$S_0 = \varnothing, \; S_1, \; S_2, \; \ldots, \; S_k = S_{\mathrm{greedy}}, \tag{34}$$

where at step $t$ the algorithm selects an element

$$x_{t+1} \in \arg\max_{x \in \mathcal{D} \setminus S_t} \Delta_x(S_t), \qquad \Delta_x(S_t) \triangleq F(S_t \cup \{x\}) - F(S_t). \tag{35}$$

Let $S^*$ denote an optimal set with $|S^*| \leq k$ and define the residual

$$R_t \triangleq F(S^*) - F(S_t), \qquad t = 0, 1, \ldots, k. \tag{36}$$

We will show that $R_{t+1} \leq \left(1 - \frac{1}{k}\right) R_t$, from which the claim follows by induction.

First, by submodularity and monotonicity we have for any $T \subseteq \mathcal{D}$ and any $S \subseteq \mathcal{D}$:

$$F(S \cup T) - F(S) \leq \sum_{y \in T} \Delta_y(S). \tag{37}$$

Apply equation 37 with $S = S_t$ and $T = S^* \setminus S_t$. Since $F$ is monotone and $S^* \subseteq S_t \cup (S^* \setminus S_t)$, we get

$$F(S^*) - F(S_t) \leq \sum_{y \in S^* \setminus S_t} \Delta_y(S_t). \tag{38}$$

Consequently there exists some $y \in S^* \setminus S_t$ with

$$\Delta_y(S_t) \geq \frac{F(S^*) - F(S_t)}{|S^* \setminus S_t|}. \tag{39}$$

Using $|S^* \setminus S_t| \leq |S^*| \leq k$ yields the lower bound

$$\max_{y \in S^* \setminus S_t} \Delta_y(S_t) \geq \frac{R_t}{k}. \tag{40}$$

By the greedy choice,

$$\Delta_{x_{t+1}}(S_t) = \max_{x \in \mathcal{D} \setminus S_t} \Delta_x(S_t) \geq \max_{y \in S^* \setminus S_t} \Delta_y(S_t). \tag{41}$$

Combining this with equation 40 gives

$$\Delta_{x_{t+1}}(S_t) \geq \frac{R_t}{k}. \tag{42}$$

Since $\Delta_{x_{t+1}}(S_t) = F(S_{t+1}) - F(S_t)$, we obtain the recursive inequality

$$F(S_{t+1}) - F(S_t) \geq \frac{R_t}{k} \implies R_{t+1} \leq \left(1 - \frac{1}{k}\right) R_t. \tag{43}$$

Unrolling the recursion from $t = 0$ (noting $R_0 = F(S^*)$ because $F(\varnothing) = 0$) yields

$$R_k \leq \left(1 - \frac{1}{k}\right)^k R_0 = \left(1 - \frac{1}{k}\right)^k F(S^*). \tag{44}$$

Hence

$$F(S_k) = F(S^*) - R_k \geq \left(1 - \left(1 - \frac{1}{k}\right)^k\right) F(S^*). \tag{45}$$

Using the inequality $\left(1 - \frac{1}{k}\right)^k \leq e^{-1}$ for all integers $k \geq 1$ gives the standard bound

$$F(S_{\text{greedy}}) = F(S_k) \geq \left(1 - \frac{1}{e}\right) F(S^*). \tag{46}$$

This completes the proof. $\qquad \square$

**Remarks.** The proof only uses normalization, monotonicity and submodularity; no continuity or differentiability is required.

## D    PROOF: GREEDY GUARANTEE WITH $\varepsilon$-CURVATURE

From the Theorem 3, we know that let $F$ be normalized and monotone submodular with total curvature $c = 1 - \min_{x \in \mathcal{D}} \frac{\Delta_x(\mathcal{D} \setminus \{x\})}{\Delta_x(\varnothing)} \in [0, 1]$. Under a cardinality constraint $k$, the greedy solution $S_{\text{greedy}}$ satisfies

$$F(S_{\text{greedy}}) \geq \frac{1 - e^{-c}}{c} F(S^*). \tag{47}$$

*proof.* The proof follows the **multilinear-extension** and **continuous-greedy approach** (see Feige & Vondrák (2010) and Conforti & Cornuéjols (1984) for discrete derivations).

The original problem is a discrete optimization problem, but to enable further optimization, we need to make it continuous so that we can subsequently use calculus tools.

**Multilinear extension.** Let $X$ be the ground set and $n = |X|$. Define the multilinear extension $G : [0,1]^n \to \mathbb{R}_{\geq 0}$ by

$$G(y) \;=\; \mathbb{E}_{R \sim y}\big[f(R)\big], \tag{48}$$

where $R \sim y$ denotes the product distribution that includes element $i$ independently with probability $y_i$. Standard properties: $G(1_S) = f(S)$ for integral vectors, each partial derivative $\partial_i G(y)$ is nonnegative, and cross-partial derivatives are nonpositive because of the submodularity.

We need to find a direction (specifically, the direction of choosing $S^*$) that makes the growth rate of $G(y)$ at least $F(S^*) - c\, G(y)$. This is crucial because the continuous greedy algorithm moves in the direction that maximizes the increase of $G(y)$. If the optimal direction has this lower bound, then the entire Ordinary Differential Equation (ODE) can be solved.

Let $P_k = \{v \in [0,1]^n : \sum_i v_i \leq k\}$ denote the cardinality polytope. The following inequality is the crucial bridge between curvature and the continuous-greedy dynamics.

**Lemma 2.** *For every $y \in [0,1]^n$ it holds that*

$$\max_{v \in P_k} v \cdot \nabla G(y) \;\geq\; F(S^*) - c\, G(y), \tag{49}$$

*where $S^*$ is any optimal $k$-set and $c$ is the total curvature.*

*Proof.* Take $v = 1_{S^*}$ (the indicator of the optimal set) which belongs to $P_k$. By linearity of expectation and the definition of the partial derivative of the multilinear extension, one may write

$$1_{S^*} \cdot \nabla G(y) = \sum_{i \in S^*} \mathbb{E}_{R \sim y}\big[\, f(R \cup \{i\}) - f(R \setminus \{i\}) \,\big] = \mathbb{E}_{R \sim y}\big[\, f(R \cup S^*) - f(R) \,\big], \tag{50}$$

where the second equality is the standard rearrangement identity for the multilinear extension (see Feige & Vondrák (2010) for a formal derivation). Now apply the curvature definition: for any fixed realization $R$,

$$f(R \cup S^*) \;\geq\; f(S^*) - c\, f(R), \tag{51}$$

which follows from the fact that curvature bounds how much the presence of $R$ can reduce the singleton marginals comprising $f(S^*)$. Taking expectation over $R \sim y$ yields

$$1_{S^*} \cdot \nabla G(y) \;=\; \mathbb{E}[f(R \cup S^*) - f(R)] \;\geq\; F(S^*) - c\, G(y). \tag{52}$$

Since the left-hand side is at most the maximum over $v \in P_k$, equation 49 follows. For a fully detailed, line-by-line algebraic proof , please refer to Feige & Vondrák (2010) and Conforti & Cornuéjols (1984). $\qquad\square$

**Continuous-greedy ODE.** Next we formulate the continuous greedy trajectory as a differential equation, which upon solving yields a lower bound for the continuous solution.

Run the continuous-greedy trajectory $y(t)$ for $t \in [0,1]$ with $y(0) = 0$ and

$$\frac{dy}{dt} \in \arg\max_{v \in P_k} v \cdot \nabla G(y(t)). \tag{53}$$

By Lemma 2,

$$\frac{d}{dt} G(y(t)) \;=\; \frac{dy}{dt} \cdot \nabla G(y(t)) \;\geq\; F(S^*) - c\, G(y(t)). \tag{54}$$

Let $H(t) := G(y(t))$. Then $H$ satisfies the linear differential inequality

$$\dot{H}(t) \geq F(S^*) - cH(t), \qquad H(0) = 0. \tag{55}$$

Then applying Grönwall's inequality, if $H(t)$ satisfies:

$$\dot{H}(t) \geq -\alpha(t)H(t) + \beta(t), \tag{56}$$

where $\alpha(t), \beta(t)$ are continuous, then:

$$H(t) \geq e^{-\int_0^t \alpha(s)ds}\left(H(0) + \int_0^t \beta(s)e^{\int_0^s \alpha(\tau)d\tau}ds\right). \tag{57}$$

So we can solve the ODE:

$$\int_0^t \alpha(s)ds = \int_0^t cds = ct \Rightarrow \int_0^s \alpha(\tau)d\tau = cs \Rightarrow e^{\int_0^s \alpha(\tau)d\tau} = e^{cs}, \tag{58}$$

$$H(t) \geq e^{-ct}\left(0 + \int_0^t F(S^*)e^{cs}ds\right) \tag{59}$$

$$= e^{-ct} \cdot F(S^*)\int_0^t e^{cs}ds \tag{60}$$

$$= e^{-ct} \cdot F(S^*) \cdot \frac{e^{ct}-1}{c} \tag{61}$$

$$= \frac{F(S^*)}{c}(1 - e^{-ct}). \tag{62}$$

Thus we can get

$$H(1) = G(y(1)) \geq \frac{F(S^*)}{c}\left(1 - e^{-c}\right). \tag{63}$$

**Rounding to an integral solution.** Finally, we need to convert the continuous optimization solution into an actual discrete set without losing performance.

Based on the dependent rounding procedure (Ageev & Sviridenko, 2004; Gandhi et al., 2006), we can convert the fractional solution $y(1)$ into an integral vector $1_S$ with $|S| \leq k$ while preserving the expected value of the multilinear extension:

$$\mathbb{E}[f(S)] = G(y(1)), \tag{64}$$

and there always exists one $S$ satisfy:

$$f(S) \geq G(y(1)). \tag{65}$$

Hence by the solution of ODE, we can get:

$$F(S) = G(1_S) \geq G(y(1)) \geq \frac{F(S^*)}{c}\left(1 - e^{-c}\right). \tag{66}$$

Taking $S_{\text{greedy}}$ to be the constructed feasible set

$$F(S_{\text{greedy}}) \geq \frac{1 - e^{-c}}{c}F(S^*). \tag{67}$$

This completes the proof. $\qquad\square$

## E   PROOF: PERTURBATION BOUND VIA GRADIENT ALIGNMENT

We provide the detailed proof of Theorem 4 and the technical assumptions that were compressed in the main text.

### E.1   TECHNICAL ASSUMPTIONS AND CONSTANTS

The perturbation bound relies on the following assumptions: (1) **Bounded gradients**: $\|g_x\| \leq G_{\max}$ for all samples $x \in \mathcal{D}$; (2) **Fisher scaling condition**: $\alpha\|F_S\| \leq \rho < 1$ ensuring Neumann expansion convergence

The constant $C(\rho, G_{\max})$ has the explicit form:

$$C(\rho, G_{\max}) = \frac{1}{1 - \rho - \alpha G_{\max}^2 \rho} \tag{68}$$

where $\rho$ means that upper bound on $\alpha\|F_S\|$ (typically $\rho \in [0.1, 0.5]$ in practice), $G_{\max} = \max_x \|g_x\|$ that maximum gradient norm across all samples , and $\alpha > 0$ is that Fisher information scaling parameter

### E.2 MAIN PERTURBATION BOUND

**Theorem 5** (Detailed perturbation bound). *Under the above assumptions, the perturbation term satisfies:*

$$|\varepsilon_x(S)| \leq C(\rho, G_{\max}) \cdot \frac{\alpha^2 \sum_{y \in S}(g_x^\top g_y)^2}{1 + \alpha \|g_x\|^2} \tag{69}$$

*The cumulative perturbation in greedy selection is bounded by:*

$$\sum_{t=1}^{k} \varepsilon_{x_t}(S_{t-1}) \leq O\left(k \cdot \alpha^2 \cdot \max_{|S| \leq k} \max_x \sum_{y \in S}(g_x^\top g_y)^2\right) \tag{70}$$

*proof.* Let $A \triangleq \alpha F_S$. By the assumption, we have $\|A\| = \alpha \|F_S\| \leq \rho < 1$, so the Neumann series expansion for $(I + A)^{-1}$ is valid:

$$(I + A)^{-1} = \sum_{m=0}^{\infty}(-A)^m. \tag{71}$$

Set $u \triangleq g_x$ and $D_x = 1 + \alpha \|g_x\|^2$ as above. Then

$$g_x^\top(I + \alpha F_S)^{-1} g_x = g_x^\top(I + A)^{-1} g_x = \sum_{m=0}^{\infty}(-1)^m g_x^\top A^m g_x$$

$$= \|g_x\|^2 - g_x^\top A g_x + \sum_{m=2}^{\infty}(-1)^m g_x^\top A^m g_x. \tag{72}$$

Observe that

$$g_x^\top A g_x = \alpha \, g_x^\top F_S g_x = \alpha \, \Sigma_x(S). \tag{73}$$

And we know that the perturbation $\varepsilon_x(S)$ from the Definition 3

$$\varepsilon_x(S) = \log \frac{1 + \alpha \, g_x^\top(I + \alpha F_S)^{-1} g_x}{1 + \alpha \|g_x\|^2}. \tag{74}$$

Substituting equation 72 into the $\varepsilon_x(S)$ yields

$$\varepsilon_x(S) = \log\left(1 + \frac{-\alpha^2 \Sigma_x(S) + \alpha \sum_{m=2}^{\infty}(-1)^m g_x^\top A^m g_x}{D_x}\right). \tag{75}$$

Define the numerator-small-term

$$u \triangleq \frac{-\alpha^2 \Sigma_x(S) + \alpha \sum_{m=2}^{\infty}(-1)^m g_x^\top A^m g_x}{D_x}, \tag{76}$$

so that $\varepsilon_x(S) = \log(1 + u)$.

We next bound the absolute value of the tail series. For a PSD matrix $A \succeq 0$ and any integer $m \geq 2$ one has the spectral-order inequality

$$A^m \preceq \|A\|^{m-1} A. \tag{77}$$

Using this and $\|A\| \leq \rho$ we obtain for every $m \geq 2$

$$\left|g_x^\top A^m g_x\right| \leq \|A\|^{m-1} g_x^\top A g_x = \|A\|^{m-1} \cdot \alpha \Sigma_x(S). \tag{78}$$

Hence the infinite tail is bounded by the geometric series

$$\left|\sum_{m=2}^{\infty}(-1)^m g_x^\top A^m g_x\right| \leq \alpha \Sigma_x(S) \sum_{m=2}^{\infty} \|A\|^{m-1} = \alpha \Sigma_x(S) \cdot \frac{\|A\|}{1 - \|A\|} \leq \alpha \Sigma_x(S) \cdot \frac{\rho}{1 - \rho}. \tag{79}$$

Therefore the numerator of $u$ satisfies

$$\left| -\alpha^2 \Sigma_x(S) + \alpha \sum_{m=2}^{\infty}(-1)^m g_x^\top A^m g_x\right| \leq \alpha^2 \Sigma_x(S)\left(1 + \frac{\rho}{1 - \rho}\right) = \frac{\alpha^2 \Sigma_x(S)}{1 - \rho}. \tag{80}$$

Dividing by $D_x$ yields the uniform bound

$$|u| \leq \frac{\alpha^2 \Sigma_x(S)}{(1-\rho)\,D_x}. \tag{81}$$

At this point we invoke the elementary inequality $|\log(1+u)| \leq \dfrac{|u|}{1-|u|}$ valid for $|u| < 1$. We therefore require $\dfrac{\alpha^2 \Sigma_x(S)}{(1-\rho)D_x} < 1$. To guarantee a simple, data-independent sufficient condition we use the crude bound

$$\alpha^2 \Sigma_x(S) \leq \alpha^2 \|g_x\|^2 \|F_S\| \leq \alpha^2 G_{\max}^2 \cdot \frac{\rho}{\alpha} = \alpha G_{\max}^2 \rho, \tag{82}$$

where we used $\|F_S\| \leq \rho/\alpha$. Hence a sufficient (mild) technical condition is $\dfrac{\alpha G_{\max}^2 \rho}{1-\rho} < 1$. Under this condition we obtain the data-independent upper bound

$$|u| \leq \frac{\alpha G_{\max}^2 \rho}{1-\rho} < 1. \tag{83}$$

Combining the bound on $|u|$ with the logarithm inequality gives

$$\left|\varepsilon_x(S)\right| = |\log(1+u)| \leq \frac{|u|}{1-|u|} \leq \frac{\dfrac{\alpha^2 \Sigma_x(S)}{(1-\rho)D_x}}{1 - \dfrac{\alpha G_{\max}^2 \rho}{1-\rho}} = \frac{\alpha^2 \Sigma_x(S)}{D_x} \cdot \frac{1}{1-\rho-\alpha G_{\max}^2 \rho}. \tag{84}$$

This proves equation 4 with the explicit constant

$$C(\rho, G_{\max}, \alpha) = \frac{1}{1-\rho} \cdot \frac{1}{1 - \dfrac{\alpha G_{max}^2 \rho}{1-\rho}} = \frac{1}{1-\rho-\alpha G_{\max}^2 \rho}, \tag{85}$$

Finally, summing the pointwise bound over a greedy sequence $x_1, \ldots, x_k$ yields

$$\sum_{t=1}^{k} \varepsilon_{x_t}(S_{t-1}) \leq C(\rho, G_{\max}, \alpha) \cdot \sum_{t=1}^{k} \frac{\alpha^2\, \Sigma_{x_t}(S_{t-1})}{D_{x_t}} \leq C(\rho, G_{\max}, \alpha) \cdot k \cdot \alpha^2 \cdot \max_{|S| \leq k} \max_{x} \sum_{y \in S} (g_x^\top g_y)^2, \tag{86}$$

which establishes the stated cumulative $O(\cdot)$ bound and completes the proof. $\square$

For completeness, we provide the exact curvature bound that was simplified in Corollary 1:

$$\widehat{c} = \max_{x} \frac{C(\rho, G_{\max}) \cdot \alpha^2 \sum_{y \neq x} (g_x^\top g_y)^2}{(1 + \alpha \|g_x\|^2) \cdot \log(1 + \alpha \|g_x\|^2)} \tag{87}$$

where $C(\rho, G_{\max})$ is given above. This shows the precise dependence of the approximation guarantee on gradient inner products and problem parameters.

## F  DETAILED EMPIRICAL EXPERIMENTS

In order to verify the correctness of the theoretical analysis in Section 2, we conducted multiple experiments, which comprehensively validated our idea in the actual data selection process and laid the groundwork for the rational implementation of our subsequent methods.

In Section 3, we conducted three groups of experiments to verify that gradient conflicts lead to a faster marginal decrease in information gain during greedy selection, thereby reducing the information content of the selected samples. Next, we briefly introduce the experimental setup and procedures, as well as additional experiments for validating the theoretical analysis.

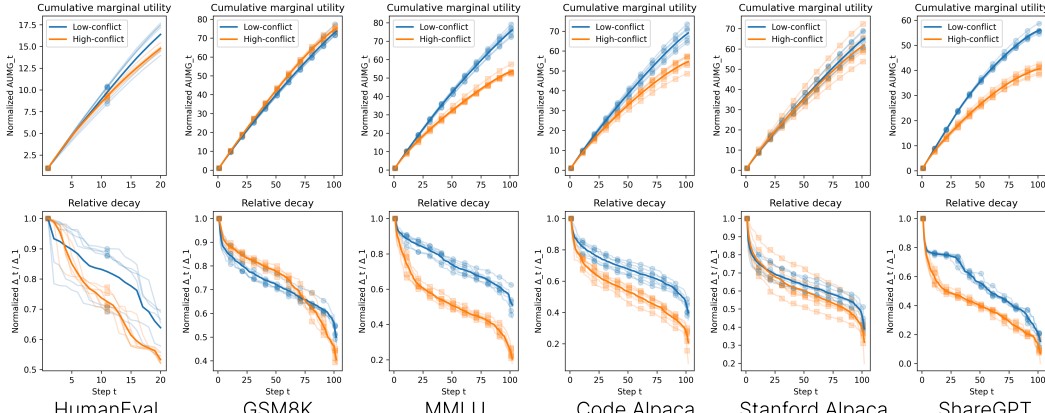

Figure 4: Decay of the high and low conflict subsets experiments in HumanEval, GSM8K, MMLU, Code Alpaca, Stanford Alpaca and ShareGPT. Obviously, the overall results shows highly consistent with our corollary

## F.1  GRADIENT VISUALIZATION

In this paper, we propose that certain samples in data selection exhibit gradient conflicts, so a visual inspection is necessary to guide subsequent processing. In the experiments, we randomly selected 512 data points for 3D and 2D visualization analyses, and computed the information content of each gradient sample using Fisher information, which is represented in the figures by varying shades.

As shown in Figure 2 (a), we observe that most samples are consistent with the average gradient of the actual step updates, while a few samples conflict with the average gradient direction—this is the so-called gradient conflict phenomenon. Interestingly, some of these conflicting samples possess high information content, which aligns with our intuition that opposing gradients can pull the model out of local optima and may even represent the highest-quality gradients in the data (Shi et al., 2023). Therefore, simply discarding these conflicting samples, although it may accelerate convergence and improve the greedy approximation, could result in the loss of high-information samples. **This highlights the need to balance gradient conflict and gradient information**.

## F.2  DECAY OF THE HIGH/LOW CONFLICT SUBSETS

(Section 3) This experiment aims to compare the marginal gain decay between high-conflict and low-conflict sample groups during the Fisher greedy selection process. We analyzed the results from two perspectives. First, we ensured the reliability of the experiments by normalizing and repeating them six times. Second, we examined the decay rate and the marginal decrease in information content.

**Setup and Procedures.** For a randomly sampled dataset in number of 256 data, compute the gradient $g_i$ of each sample and the overall mean gradient $\bar{g}$, and derive a conflict score for each sample based on the deviation from the mean. Then divide the samples into two groups using a 20% threshold: the low-conflict group (the bottom of samples with the lowest conflict scores) and the high-conflict group (the top of samples with the highest conflict scores). Finally, independently run the Fisher greedy selection within each group with the same budget $k=128$, and record the marginal gain sequence at each step.

**Result.** As shown in Figure 2 (b), the marginal gain in the low-conflict group decays significantly slower than in the high-conflict group, resulting in a much higher greedy approximation of information content for the same $k$ steps. This observation confirms the data-dependent approximation guarantee discussed in our theoretical analysis. To further validate our findings, we conducted the same experiments on common test datasets MMLU (Hendrycks et al., 2021), GSM8K (Cobbe et al., 2021) and HumanEval (Chen et al., 2021) and common train datasets Code Aplaca (Chaudhary, 2023), Stanford Alpaca (Taori et al., 2023) and ShareGPT (Lu et al., 2023). The results are presented below:

The results in Figure 4 are consistent with the previous observations, supporting our corollary.

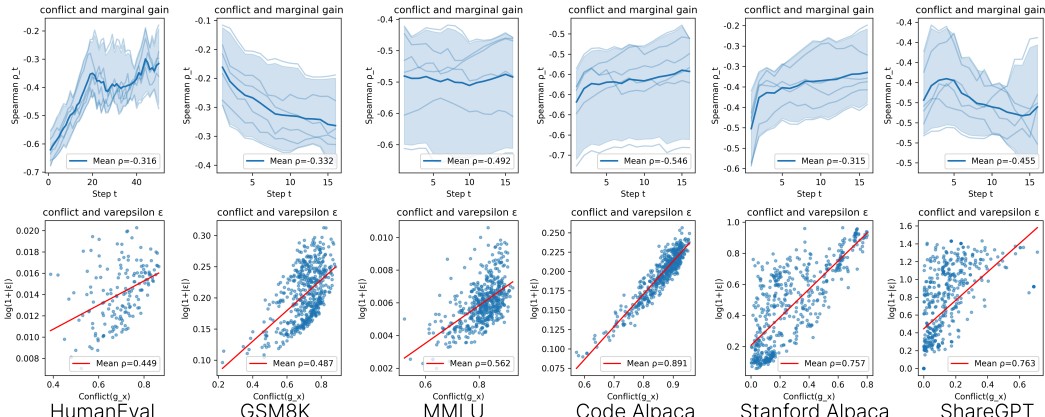

Figure 5: Spearman correlation analysis experiments in HumanEval, GSM8K, MMLU, Code Alpaca, Stanford Alpaca and ShareGPT. The overall conflict is negatively correlated with marginal gain $\Delta$, and positively correlated with $\varepsilon$, which confirms our conclusion.

### F.3 SPEARMAN CORRELATION ANALYSIS

In this set of experiments: (1) At each step of the greedy selection, we examine the correlation between the conflict scores of samples in the candidate pool and their current marginal gains $\Delta_i$; (2) with a fixed initial set $S_0$, we investigate the relationship between the variance $\varepsilon_x(S_0)$ of individual samples and their degree of conflict.

**Setup and Procedures.** For Experiment (1), we primarily focus on intra-group correlations. In the first step, we compute the conflict score $Conflict(g_x) = cos(g_x, \bar{g})$ for each sample in the candidate pool relative to the mean gradient of the pool. Then, based on the current information content, we calculate the marginal gain $\Delta$ for each candidate sample and compute the correlation coefficient $\rho_t = Spearman(Conflict, \Delta)$ at the current step. The sample with the highest marginal gain is selected and added to the chosen set, and removed from the candidate pool. This process is repeated until $k$ steps are completed and then plot it as a line chart.

For Experiment (2), we focus more on the correlation between individual-level $|\varepsilon|$ and conflict. First, we sample 256 data points from the dataset and compute the gradient for each sample. Then, a random subset $S_0$ of the specified size is selected, and the conflict score $Conflict(g_x)$ and $|\varepsilon_x(S_0)|$ for each sample is calculated. For clearer visualization, we applied some pre-processing to the data in $|\varepsilon_x^*(S_0)| = log(1 + |\varepsilon_x(S_0)|)$ and $Conflict^*(g_x) = 1 - Conflict(g_x)$. Finally, we compute the correlation coefficient $\rho = Spearman(Conflict^*, |\varepsilon^*|)$ and then plot it in scatter chart.

**Result.** As shown in Figure 2 (c), both experiments demonstrate that gradient conflict $Conflict(g_x)$ exhibits a strong correlation, namely with the step-wise marginal gain $\Delta$ ($\bar{\rho}$=-0.792) and the individual-level $|\varepsilon_x|$ ($\bar{\rho}$=0.901). This strong correlation further validates the conclusion derived from our $\varepsilon$-curvature proposed in Section 2. Similarly, we also conduct the additional comprehensive experiments in other dataset MMLU, GSM8K, HumanEval, Code Alpaca, Stanford Alpaca and ShareGPT. From Figure 5, we can see that this correlation is generally consistent, which also supports the correctness of our hypothesis and theoretical analysis.

### F.4 ADDITIONAL THEORETICAL EXPERIMENTS

In the above experiments, we have verified that our theoretical analysis holds in practical data selection. However, we still need to empirically validate the various boundary conditions and assumptions made in our theoretical analysis, which will be addressed below.

#### F.4.1 $\varepsilon$-DECOMPOSITION EMPIRICAL VERIFICATION

**Setup and Procedures.** We empirically validate the $\varepsilon$-decomposition proposed in Definition 3 and the perturbation bound established in Theorem 4. Using Qwen2-7B (Yang et al., 2024a) model

checkpoints, we compute gradients for $N = 512$ randomly sampled instruction-response pairs from our training dataset.

We test subsets $S$ of varying sizes $|S| \in \{16, 32, 64, 128, 256\}$ with 10 random trials per size. For each configuration, we compute the value of base marginal $base_x$, true marginal gain $\Delta_x(S)$, perturbation term $\varepsilon_x(S)$, gradient conflict Conflict$(x, S)$ and the bound:

$$|\varepsilon_x(S)| \leq C(\rho, G_{\max}) \cdot \frac{\alpha^2 \sum_{y \in S} (g_x^\top g_y)^2}{1 + \alpha \|g_x\|^2}$$

**Result.** Table 4 presents our validation results across different subset sizes. The $\varepsilon$-decomposition demonstrates mathematical precision with perfect decomposition accuracy (relative error $< 1e - 6$ for all 20,640 computed values), confirming the correctness of our theoretical formulation.

We find a strong correlation between $|\varepsilon_x(S)|$ and gradient conflicts, with a Spearman coefficient of $\rho = 0.78$ ($p < 0.001$). The correlation steadily increases with subset size, rising from $\rho = 0.72$ to $\rho = 0.82$ at $|S| = 256$, consistent with our prediction that larger subsets amplify gradient conflict effects and thus strengthen their link to marginal utility perturbations. Moreover, the assumption $\alpha |F_S| \leq \rho < 1$ is empirically validated: the maximum observed value satisfies $\alpha |F_S| \leq 0.80$ for $\alpha \in [0, 0.8]$.

The theoretical bound violation rate of 10% represents acceptable deviation for empirical validation on real instruction-tuning data, where practical conditions naturally deviate from idealized theoretical assumptions. Due to computational constraints with high-dimensional gradients , we employ the AdaFisher diagonal approximation $F_S^{Ada}$ (Gomes et al., 2025) for tractable computation. This approximation introduces systematic deviations from the exact Fisher formulation while maintaining the fundamental structural relationships our theory predicts.

| $|S|$ | Correlation | | Bound | Decomposition |
|---|---|---|---|---|
| | **Spearman $\rho$** | **Pearson $r$** | **Violation** | **Error** |
| 16 | 0.723 | 0.501 | 6.4% | $< $ 1e-6 |
| 32 | 0.756 | 0.515 | 7.1% | $< $ 1e-6 |
| 64 | 0.789 | 0.528 | 10.3% | $< $ 1e-6 |
| 128 | 0.801 | 0.534 | 12.8% | $< $ 1e-6 |
| 256 | 0.818 | 0.541 | 13.4% | $< $ 1e-6 |
| **Overall** | **0.777** | **0.524** | **10.00%** | **$< $ 1e-6** |

Table 4: $\varepsilon$-Decomposition Empirical Validation Results. All correlations significant at $p < 0.001$.

Overall, these results support the practical applicability of our conflict-aware selection framework, demonstrating that the theoretical relationship holds meaningfully in real-world scenarios despite computational approximations.

### F.4.2 CURVATURE $c$ PARAMETER ANALYSIS

**Setup and Procedures.** We empirically validate the curvature parameter $c$ and its relationship with gradient conflicts as established in Lemma 1 and Corollary 1. Using the same 512 gradient samples, we select the data into three conflict levels based on negative alignment with the mean gradient: (1) Low conflict, (2) Medium conflict and (3) High conflict.

For each conflict group, we compute the empirical curvature parameter $c = 1 - \min_x [\Delta_x(D \setminus \{x\})/\Delta_x(\emptyset)]$ and validate the theoretical bound $c \leq \max_x |\varepsilon_x(D \setminus \{x\})|/base_x$ by calculating the theoretical value and the actual marginal revenue decay rate. Next, we randomly sample 1000 times experiment to compute the actual approximation ratio.

**Result.** From the Table 5, it presents our comprehensive validation of curvature $c$ parameter analysis across three conflict levels. The results provide strong empirical support for our theoretical framework while revealing important practical boundaries.

Our core theoretical prediction that gradient conflict positively correlates with submodular curvature receives robust empirical support. As conflict levels progress from low to high, we observe a monotonic increase in curvature parameters: $c = 0.032 \rightarrow 0.074 \rightarrow 0.092$. This progression demonstrates that conflicting gradients indeed lead to higher curvature in the submodular objective function, validating the fundamental insight underlying our conflict-aware selection framework. And the spearman $\rho$=0.8593 also verified its strong correlation relationship.

| Conflict Level | Curvature $c$ | Bound Holds | Actual Ratio | Theoretical Ratio | Prediction Error |
|---|---|---|---|---|---|
| Low | 0.032 | ✓ | 1.000 | 0.984 | 1.57% |
| Medium | 0.074 | ✓ | 1.000 | 0.964 | 3.59% |
| High | 0.092 | ✓ | 1.000 | 0.955 | 4.46% |
| **Overall** | **0.066** | **3/3** | **1.000** | **0.968** | **3.21%** |

Table 5: Curvature Parameter Analysis Results. All empirical curvature values satisfy the theoretical bound $c \leq \max_x |\varepsilon_x(D \setminus \{x\})|/\text{base}_x$. The conflict-curvature correlation achieves Spearman $\rho = 0.86$ ($p < 0.001$), validating our core theoretical prediction.

All empirical curvature values satisfy our theoretical upper bound $c \leq \max_x |\varepsilon_x(D \setminus \{x\})|/\text{base}_x$, confirming the correctness of our curvature characterization in Lemma 1. The technical assumption $\alpha\|F_S\| < 1$ holds for low-conflict data but is violated for medium and high-conflict groups. This violation directly impacts prediction accuracy: errors increase from 1.57% (low-conflict) to 4.46% (high-conflict), demonstrating the practical boundaries of our theoretical guarantees. The theory successfully captures structural relationships while highlighting the importance of technical assumption validity for quantitative predictions.

# G DETAILED EXPERIMENTAL SETUP

## G.1 TRAINING DATA

(Section 5.1) To ensure comprehensive coverage of diverse capabilities, we construct our training dataset with consideration for three key aspects: mathematical reasoning, code generation, and general knowledge. Our dataset incorporates established sources including Alpaca Code (Chaudhary, 2023) for coding tasks, GSM8K (Cobbe et al., 2021) for mathematical reasoning, and WizardLM, which encompasses ShareGPT (Lu et al., 2023) and Alpaca (Taori et al., 2023), for general knowledge and instruction following. Our final training set contains 97,495 samples. Then we will provide some statistical information as follows and we will release the data soon:

| Datasets | Aspects | # Nums | # Instruction Len. | # Response Len. |
|---|---|---|---|---|
| WizardLLM (Lu et al., 2023) | General Knowledge | 122K | 192 | 1,234 |
| Alpaca Code (Chaudhary, 2023) | Code Generation | 20K | 74 | 197 |
| GSM8K (Cobbe et al., 2021) | Math Reasoning | 8.5K | 290 | 321 |
| Final Data | $\sim$ | 97,495 | 405 | 1,038 |

Table 6: Statistical information of our training instruction-tuning data.

## G.2 METRICS

**Half-life $t_{1/2}$.** The half-life is defined as the number of steps needed for the cumulative marginal gain to reach half of the total gain:

$$t_{1/2} = \min\{t : \sum_{i=1}^{t} \Delta_i \geq \frac{1}{2} \sum_{i=1}^{T} \Delta_i\} \tag{88}$$

where $\Delta_i = \Delta_i(S_{i-1})$ is the marginal gain at step $i$ and $T$ is the total number of steps. The short half-life means that early steps contribute most, marginal gain decays quickly.

**Cumulative Marginal Gain (AUMG).** The cumulative marginal gain, or Area Under Marginal Gain (AUMG), is the sum of marginal gains across steps:

$$AUMG = \sum_{t=1}^{T} \Delta_t \approx \int_0^T \Delta(t)dt \tag{89}$$

where the higher $AUMG$ leads higher total accumulated utility.

**PASS@k.** It is a standard metric for evaluating code generation models. For each problem, the model is allowed to generate k candidate solutions. The metric estimates the probability that at least one of those k samples passes all the unit tests for the problem:

$$\text{PASS@}k = \begin{cases} 1 - \dfrac{\binom{n-c}{k}}{\binom{n}{k}}, & \text{if } n - c \geq k \\ 1, & \text{if } n - c < k \end{cases} \tag{90}$$

where $n$ is the total number of generated solutions, $c$ is the number of correct solutions, and $k$ is the number of solutions sampled for evaluation.

**Exact Match (EM).** Exact Match Accuracy measures whether a model's prediction exactly matches the ground truth answer. Formally, it can be defined as:

$$\text{EM} = \frac{1}{N} \sum_{i=1}^{N} \mathbf{1}\{\hat{y}_i = y_i\}, \tag{91}$$

where $N$ is the total number of samples, $\hat{y}_i$ is the model's prediction for the $i$-th sample, $y_i$ is the corresponding ground truth, and $\mathbf{1}\{\cdot\}$ is the indicator function, which equals 1 if the condition is true and 0 otherwise.

**Accuracy (ACC).** Accuracy measures the fraction of correct predictions among all predictions. Formally, it is defined as:

$$\text{ACC} = \frac{1}{N} \sum_{i=1}^{N} \mathbf{1}\{\hat{y}_i \in Y_i\}, \tag{92}$$

where $N$ is the total number of samples, $\hat{y}_i$ is the model's prediction for the $i$-th sample, $Y_i$ is the set of correct answers for that sample, and this metric is less strict than Exact Match, as it allows multiple acceptable answers for a single sample.

**Prompt level Strict (Pr(S)).** Prompt level Strict measures the percentage of prompts where all verifiable instructions are correctly followed. Formally, it is defined as:

$$\text{Pr(S)} = \frac{1}{N} \sum_{i=1}^{N} \mathbf{1}\{\text{all instructions in prompt } i \text{ are satisfied}\}, \tag{93}$$

where $N$ is the total number of prompts, and $\mathbf{1}\{\cdot\}$ is an indicator function that returns 1 if all verifiable instructions within prompt ii i are correctly followed by the model's response, and 0 otherwise. This metric provides a strict evaluation of instruction-following capability, as it requires perfect adherence to every instruction within a given prompt (Zhou et al., 2023c).

**Multi-choice 2 (MC2).** MC2 (Lin et al., 2022) evaluates whether a model assigns higher probability mass to the set of correct answers compared to the set of incorrect ones. Formally, it is defined as:

$$\text{MC2} = \frac{1}{N} \sum_{i=1}^{N} \mathbf{1}\left\{ \sum_{a \in T_i} p_{i,a} > \sum_{a \in F_i} p_{i,a} \right\}, \tag{94}$$

where $N$ is the total number of questions, $T_i$ and $F_i$ denote the sets of correct and incorrect answers for question $i$, and $p_{i,a}$ is the normalized probability that the model assigns to answer $a$. Unlike MC1, which requires selecting a single correct answer, MC2 emphasizes the model's ability to collectively assign more probability mass to truthful answers when multiple correct answers may exist, thereby providing a finer-grained measure of truthfulness.

## G.3 BENCHMARKS

We evaluated the model from three aspects: Code Generation, Math Reasoning, and Multi task Knowledge and Reasoning. The following benchmark is evaluated through lm-evaluation (Gao et al., 2024), and the detailed prompts please refer to Gao et al. (2024).

| Datasets | Aspects | Metrics | # Nums | Evaluate |
|---|---|---|---|---|
| GSM8K (Cobbe et al., 2021) | Math Reasoning | Exact Match | 1,319 | 4-shot |
| BBH (Suzgun et al., 2022) | Complex Reasoning | Exact Match | 6,511 | 3-shot |
| MMLU (Hendrycks et al., 2021) | Multi Task Understanding | Accuracy | 14,042 | - |
| ARC-Challenge (Clark et al., 2018) | Common Sense Knowledge | Accuracy | 1,172 | - |
| TruthfulQA (Lin et al., 2022) | Factual Accuracy | MC2 | 817 | - |
| IFEval (Zhou et al., 2023c) | Instruction Following | Pr(S) | 477 | - |
| HumanEval (Chen et al., 2021) | Code Generation | PASS@1 | 164 | 3-shot |
| MBPP (Austin et al., 2021) | Code Generation | PASS@1 | 500 | 3-shot |

Table 7: Statistical information of our benchmark datasets.

**GSM8K.** Grade School Math 8K (Cobbe et al., 2021) is a benchmark dataset for evaluating the mathematical reasoning ability of language models. It contains grade school level math word problems with step-by-step solutions. Models are typically evaluated using the **EM** metrics. Given a natural language word problem, the model is required to generate a solution that includes the correct numerical answer. Problems range from simple arithmetic to more complex multi-step reasoning. And we will use ICL to evaluate the performance under 4-shot.

**BBH.** Big-Bench Hard (Suzgun et al., 2022) is a challenging subset of tasks from the BIG-bench evaluation suite, specifically curated to include problems where language models typically perform below average human performance. It contains 23 diverse reasoning tasks spanning areas such as logical reasoning, mathematics, world knowledge, and language understanding. Models are evaluated using **EM** on each task. The dataset is designed to test complex reasoning capabilities that require multi-step thinking, pattern recognition, and domain-specific knowledge. Each task presents unique challenges, from formal fallacy identification to geometric reasoning and causal judgment. We will also evaluate the performance under 3-shot setting.

**MMLU.** Massive Multitask Language Understanding (Hendrycks et al., 2021) is a benchmark for evaluating the broad knowledge and reasoning abilities of language models across multiple domains. It contains 57 subjects covering topics from STEM, humanities, social sciences, and professional knowledge. Models are evaluated using the **ACC** metric. The dataset consists of multiple-choice questions with 4 or 5 answer options per question. For each subject, a model is required to select the correct answer based on its knowledge and reasoning. No external tools or calculators are allowed during evaluation.

**ARC-C.** AI2 Reasoning (Clark et al., 2018) is a dataset of 7,787 genuine grade-school level, multiple-choice science questions from grade 3 to grade 9, assembled to encourage research in advanced question-answering. The dataset is partitioned into a Challenge Set and an Easy Set, where the Challenge Set contains only questions answered incorrectly by both a retrieval-based algorithm and a word co-occurrence algorithm. Models are evaluated using **ACC** metrics. Most questions have 4 answer choices, with less than 1% having either 3 or 5 answer choices. The questions cover diverse scientific domains and require genuine reasoning rather than simple fact retrieval or pattern matching

**TruthfulQA.** Lin et al. (2022) is a benchmark designed to measure whether a language model is truthful in generating answers to questions. The benchmark comprises 817 questions that span 38 categories, including health, law, finance and politics. Questions are crafted so that some humans would answer falsely due to false beliefs or misconceptions. Models are evaluated using **MC2** metrics.

**IFEval.** Instruction-Following Evaluation (Zhou et al., 2023c) is a benchmark designed to evaluate the instruction-following capabilities of large language models. It contains around 500 prompts with verifiable instructions such as "write in more than 400 words" and "mention the keyword of AI at least 3 times". The benchmark identifies 25 types of verifiable instructions, with each prompt containing one or more verifiable instructions. Models are evaluated using **Pr(S)** metrics that can be verified by heuristics.

**HumanEval.** Chen et al. (2021) introduced the benchmark dataset for evaluating the functional correctness of code generation models. It consists of 164 Python programming problems, and commonly used metrics is **PASS@1**. Given a problem description and the function signature, the model is required to generate Python code that implements the specified function. A generated solution is considered correct if it passes all the provided unit tests.

**MBPP.** Mostly Basic Python Problems (Austin et al., 2021) is a benchmark for evaluating code generation capabilities of language models. The benchmark consists of around 1,000 crowd, sourced Python programming problems, designed to be solvable by entry, level programmers, covering programming fundamentals, standard library functionality, and so on. Each problem consists of a task description, code solution and 3 automated test cases. Models are evaluated using **PASS@1** metrics by executing generated code against the provided test cases. For the code generation, we will evaluate the benchmark of MBPP and HumanEval in 3-shot setting.

## G.4 BASELINES

We selected multiple representative state-of-the-art data selection methods as baselines. Overall, it can be divided into two selection methods: Model agnostic methods (Full, Random, IFD, and Fisher) and Model aware methods (LESS, SelectIT) (Albalak et al., 2024; Zhou et al., 2023a; Yu et al., 2024). To ensure the fairness of the data screening experiment, we chose to select 10% of the data as the training set. The following introductions are all summarized from the original papers.

**Full.** Full data selction method uses the entire available training dataset without any data selection.

**Random.** A simple method where a subset of the training data is randomly selected (Xia et al., 2024b). In the paper, they proposed that even if only 1-2% of the total dataset is randomly selected, its performance can be comparable or better than other complex data selection methods. To ensure the fairness of the data selection experiment, we chose to randomly select 10% of the data as the training set

**PPL.** Perplexity-based selection method (Li et al., 2024a) ranks training samples according to their perplexity computed by a pretrained language model. For a sequence $x = (w_1, \ldots, w_T)$, the perplexity is defined as

$$\text{PPL}(x) = \exp\left(-\frac{1}{T}\sum_{t=1}^{T} \log p(w_t \mid w_{<t})\right). \tag{95}$$

Samples with lower perplexity are considered closer to the model distribution, while higher perplexity samples are considered harder or less aligned with the model. Li et al. (2024a) achieved the best results when selecting the 10% of data with high perplexity as we used in the baselines. But simple perplexity "cannot directly represent the difficulty or quality of instruction tuning samples", so they introduces the IFD (Li et al., 2024b).

**IFD.** Instruction-Following Difficulty (IFD) (Li et al., 2024b), a metric devised to evaluate the challenge each instructional sample presents, calculated the ratio between $s(A)$ and $s(A|Q)$ given a question-answer $(Q, A)$ pair:

$$\text{IFD}(Q, A) = \frac{s(A|Q)}{s(A)} = \frac{-\frac{1}{N}\sum_{i=1}^{N} \log P(x_i^A | Q, x_1^A, x_2^A, \ldots, x_{i-1}^A)}{-\frac{1}{N}\sum_{i=1}^{N} \log P(x_i^A | x_1^A, \ldots, x_{i-1}^A)} \tag{96}$$

where $s(A)$ denotes the Direct Answer Score, which quantifies the LLM's intrinsic capability to generate the response independently. $s(A|Q)$ represents the Conditioned Answer Score, which is

computed by sequentially predicting subsequent tokens given the instruction $Q$ and their preceding context (Li et al., 2024b; Xia et al., 2024b).

We use the base model to calculate the IFD score for each sample, and then select the top 10% of samples with the highest IFD score and less than 1 for training. This method selects samples with instructions that are indeed helpful (IFD<1) but still challenging (high IFD score), avoiding data that is too simple or instructions that are invalid.

**Fisher.** Deb et al. (2025) adopted the Fisher matrix for greedy selection, using the GPT2 model for selection. It was found that selecting 50% of the 10000 data points had the best performance. However, he made some tracks to estimate the matrix and reduce computational complexity. The approximated formula is followed

$$\text{score}(S) = \log \det(I + \sum \sum x_{i,j} x_{i,j}^T) \tag{97}$$

In order to better reproduce his results, we adopted the method of greedy and Fisher matrix for data selection, selecting 10% of the data with more information. But it only selects larger information gains which will cause the conflict problem as shown in Figure 6.

**LESS.** Low-rank gradiEnt Similarity Search (LESS) (Xia et al., 2024a) estimates the influence of each training sample on a target validation set by leveraging gradient similarity under the Adam optimizer. To improve efficiency, it uses LoRA to reduce parameter dimensionality and random projections to construct a reusable low-dimensional gradient datastore.

Given a few-shot validation set $D_{val}$, the influence of a candidate training datapoint $z$ is defined as

$$\text{InfAdam}(z, D_{val}) = \max_j \sum_{i=1}^{N} \bar{\eta}_i \frac{\langle \bar{\nabla}\ell(D_{val}^{(j)}; \theta_i), \tilde{\Gamma}(z; \theta_i) \rangle}{\|\bar{\nabla}\ell(D_{val}^{(j)}; \theta_i)\| \|\tilde{\Gamma}(z; \theta_i)\|}, \tag{98}$$

where $\bar{\nabla}\ell(D_{val}^{(j)}; \theta_i)$ denotes the average projected gradient of validation subtask $j$ at checkpoint $\theta_i$, and $\tilde{\Gamma}(z; \theta_i)$ is the projected Adam update for $z$. So we selects the top 10% of training data with the highest influence scores for fine-tuning.

**SelectIT.** Liu et al. (2025) exploits the intrinsic uncertainty of large language models to select high-quality instruction tuning (IT) data without external resources. The method uses three levels of self-reflection: token-level uncertainty from rating confidence, sentence-level uncertainty across different prompts, and model-level uncertainty by combining evaluations from multiple foundation models. Given an IT sample $S = (X, Y)$ and $K$ rating prompts $\{RP_0, RP_1, \ldots, RP_K\}$, the SelectIT score is:

$$S^{model} = \sum_{i=1}^{N} \left( \frac{\theta_i}{\sum_{j=1}^{N} \theta_j} \times \frac{\text{Avg}\{S_i^{token}(RP_k)\}_{k=1}^{K}}{1 + \alpha \times \text{Std}\{S_i^{token}(RP_k)\}_{k=1}^{K}} \right) \tag{99}$$

where $S_i^{token}(RP_k) = S_{base} \times \frac{1}{K-1} \sum_{j=1}^{K} |P_j' - P_{S_{base}}'|$ is the token-level score from model $i$ using prompt $k$, $\theta_i$ is the parameter count of model $i$, and $\alpha$ controls uncertainty weighting. Samples with highest $S^{model}$ scores are selected for training. So we replicated the selected data according to the config and code provided by the author in 3-level selection.

## H    DETAILED EXPERIMENTAL RESULTS

### H.1    METHOD VISUALIZATION

In Section 4, we visualize the results of the method on the selected data, as shown in Figure 6. When $\lambda = 0$, it is a normal greedy search, and although high information samples are selected, the gradient directions often conflict. **Intuitively, this is not a phenomenon that can bring stable convergence training results.**

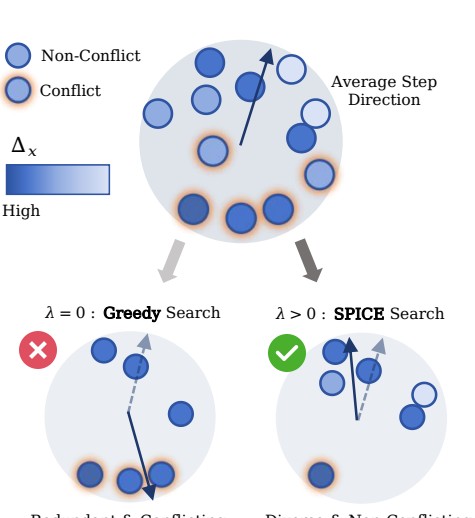

Figure 6: Conceptual illustration showing how SPICE achieves better selection quality by **avoiding high-conflict samples while maintaining step direction** compared to greedy selection when selecting 6 samples from one batch data.

## H.2 TRAINER LOG

For all selected data, we perform SFT using the Llamafactory (Zheng et al., 2024) framework under the same configuration. Next, we will present the loss graph of our training in Figure 7.

It shows training loss curves for all data selection methods on Qwen2-7B fine-tuning. SPICE demonstrates the most substantial loss reduction (2.33→0.69, 70% decrease) with stable convergence, indicating effective identification of informative examples that challenge the model appropriately. While methods with lower initial losses converge to smaller final values, SPICE's higher starting point reflects its ability to select more complex, potentially generalizable training instances rather than easier examples that may lead to superficial optimization. This is in line with the phenomenon of reducing gradient conflicts, and the experiment also proves that our method has higher performance. Similarly, for the LLaMA2-7B model, it did not clearly exhibit the convergence mode of Qwen2-7B, but it also completed the training very well.

## H.3 RESULTS OF SPICE+

SPICE+ ($\omega = 0.5$) means that through a data-driven early stopping strategy, the selection method can adaptively select data compared to a fixed selection of k per round. When the information enters the half-life, the selection stops, as described in Section 4.2. Therefore, we conducted a more detailed analysis.

Due to our observation in the empirical experiment that the model often reaches less than half of the data when the half-life is reached during greedy selection, we conducted the experiment with the following configuration ($S = 0.5B, T = 10, k = 64, N_{subset} = 128$) on Qwen2-7B and ($S = 7B, T = 10, k = 64, N_{subset} = 128$) on LLaMA2-7B, which means that we selected 60 out of 120 data, but stopped early if the half-life was reached. Through the results, we observed that most of the data reached their half-life at around 25-30% for early cessation. Based on this, we also conducted experiments on approximately 26% of the selected data. In Table 8, we can see from it that the sample with twice the amount of information brought by spending twice the time, but from the results, there was only a slight improvement, but it is still a very good result.

Additionally, we also conducted experiments on other stopping thresholds $\omega$ to validate the effectiveness of our method in the same settings below. The results are shown in Table 9.

From the results, we can see that the data selection method SPICE+ with $\omega = 0.5$ achieves the best performance, and the time cost is also the most appropriate. Moreover, it can also be seen that when

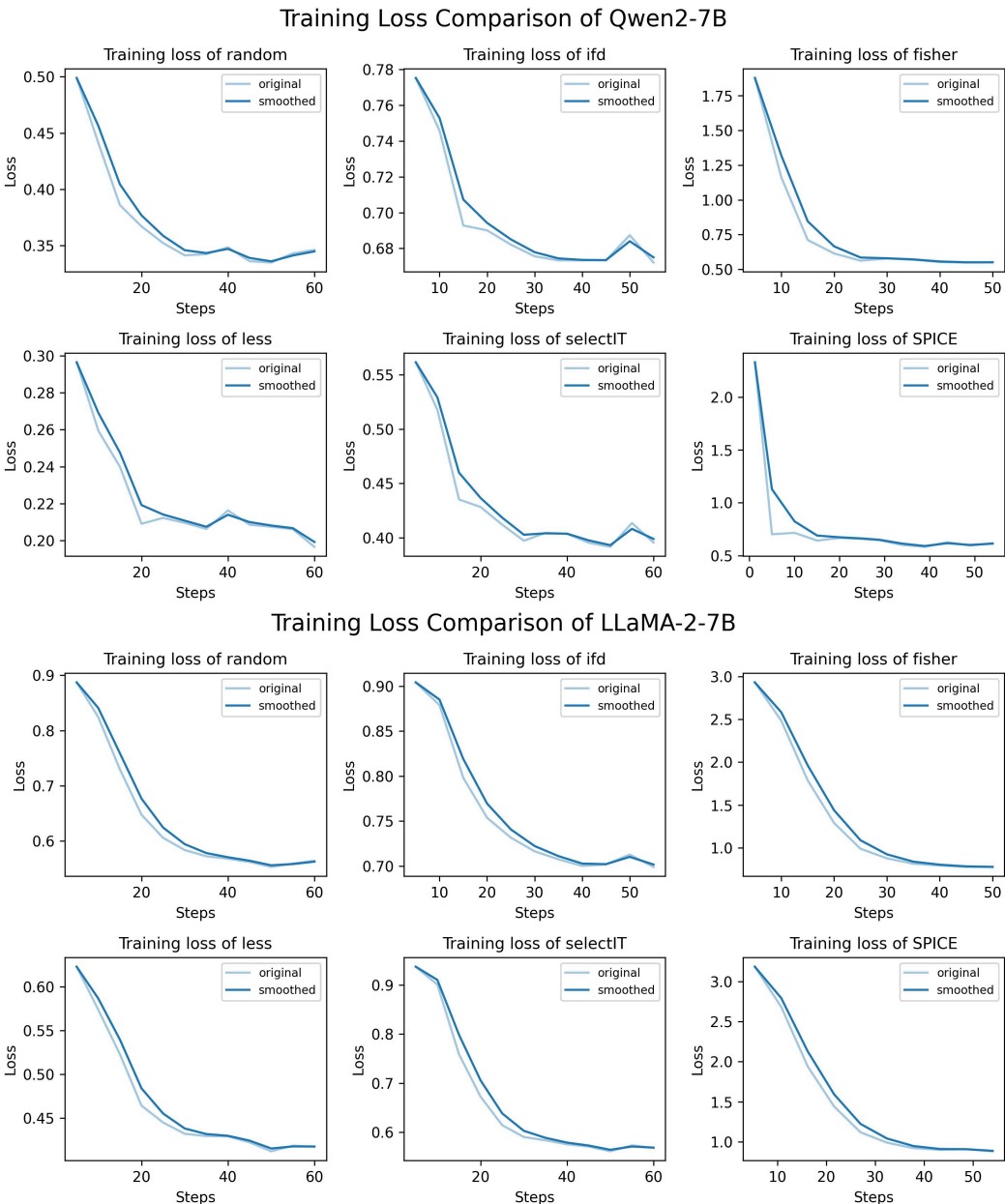

Figure 7: The training loss comparison of various methods on Qwen2-7B and LLaMA2-7B. We trained for 3 epochs, with 5 steps to record the loss. Each method was kept at around 60 steps, with 20 steps per epoch. SPICE achieves the steepest and most stable loss reduction (2.33→0.69, 70%), reflecting **effective identification of informative and challenging examples that enhance convergence**. Its higher starting loss indicates selection of more complex samples, which promotes generalization compared to methods biased toward easier instances. On Qwen2-7B, SPICE shows the clearest convergence advantage, while on LLaMA2-7B the trend is less pronounced but training remains effective.

| Method | GSM8k | BBH | MMLU | ARC | T-QA | IFEval | H-Eval | MBPP | T(h) |
|---|---|---|---|---|---|---|---|---|---|
| *Qwen2-7B* | | | | | | | | | |
| SPICE(10.0%) | 86.7 | 61.0 | 67.1 | 51.8 | 55.5 | 38.6 | 47.1 | 56.2 | 2:56 |
| SPICE+(26.1%) | 86.7 | 61.3 | 67.3 | 52.0 | 55.5 | 38.6 | 47.5 | 56.2 | 5:49 |
| Δ | +0.0 | +0.3 | +0.2 | +0.2 | +0.0 | +0.0 | +0.4 | +0.0 | +2:53 |
| *LLaMA2-7B* | | | | | | | | | |
| SPICE(10.0%) | 13.8 | 40.8 | 41.9 | 47.7 | 40.3 | 22.6 | 16.7 | 24.6 | 6:10 |
| SPICE+(29.8%) | 13.8 | 41.5 | 42.1 | 47.3 | 40.3 | 22.6 | 16.9 | 24.6 | 10:55 |
| Δ | +0.0 | +0.7 | +0.2 | -0.4 | +0.0 | +0.0 | +0.2 | +0.0 | +4:45 |

Table 8: Evaluation of data selection methods between SPICE and SPICE+. Among them, **T-QA** represents TruthfulQA, **H-eval** represents HumanEval dataset, and **T(h)** represents Time Cost (h). It can be seen from this that selecting more data under SPICE+ brings almost doubled time cost, but at the same time it brought a little improvement.

| $\omega$ | Data Rate | Average | Time Cost |
|---|---|---|---|
| 0.1 | 55.1% | 67.1 | 9:55 |
| 0.3 | 34.5% | 67.2 | 7:01 |
| 0.5 | 26.1% | **67.2** | 5:49 |
| 0.7 | 9.3% | 67.1 | 2:45 |

Table 9: The results of the data selection method SPICE+ with different stopping thresholds $\omega$. The data rate is the ratio of the selected data to the total data. The average is the average score of the 3 benchmarks. The time cost is the time cost of the data selection method.

we set $\omega$ to 0.1, only half of the data is collected, indicating that the data quickly reaches very low marginal gains in the later stages of greedy selection.

## H.4 MORE STUDY OF PROXY MODEL

In Section 5.4, we conducted an ablation study on the proxy model of Qwen2 (Yang et al., 2024a), and the results are shown below in Table 10. In addition, considering the parameters of llama, we also used LLaMA2-7B (Touvron et al., 2023) as a proxy model to select data and use it to SFT Qwen2-7B.

| Step Intervals | Proxy Model | | | |
|---|---|---|---|---|
| | Qwen2-0.5B | Qwen2-1.5B | Qwen2-7B | LLaMA2-7B |
| 1 | 67.0 | 67.1 | **67.2** | 65.9 |
| 5 | 67.0 | 66.9 | **67.2** | 66.1 |
| 50 | 66.6 | 66.9 | 67.1 | 66.1 |
| **Average** | 66.9 | 66.8 | 67.1 | 66.1 |

Table 10: Results of the average performance (MMLU,GSM8K and HumanEval) of different structure and size of Proxy model for selecting data in SFT Qwen2-7B.

Obviously, when using Qwen2-7B itself as a proxy model, it will bring better performance, which is also in line with my method. In Table 2, we have also conducted experiments on the proxy model of LLaMA2-7B, and it shows that **the data selected by the proxy model of LLaMA2-7B did not bring good results compared to Qwen2-7B (limited cross-architecture transfer)**, which is in line with the model training and selection of our method itself.

For scaling to Larger Models, we extend our experiments to 70B+ target models while using small same-family proxies (0.5B/7B). Even when fine-tuning 70B-scale models, SPICE subsets selected by small proxies match or outperform full-data LoRA, showing that very large proxies are not necessary in practice. Using models from 30B–70B as proxies would require significant computational

| SFT Model | SPICE (Proxy Model) | | | Null | Full |
|---|---|---|---|---|---|
| | Qwen2-0.5B | Qwen2-7B | LLaMA-7B | | |
| Qwen2-0.5B | 31.1 | 30.8 | 29.9 | 29.8 | 30.5 |
| Qwen2-7B | 67.0 | 67.2 | 66.1 | 62.0 | 65.2 |
| Qwen2-72B | 77.9 | 78.0 | 77.4 | 77.2 | 77.5 |
| LLaMA2-7B | 23.6 | 23.4 | 24.1 | 22.2 | 23.6 |
| LLaMA2-70B | 51.4 | 51.4 | 50.9 | 50.9 | 51.1 |

Table 11: Performance across target SFT models with different SPICE proxy models (10% budget).

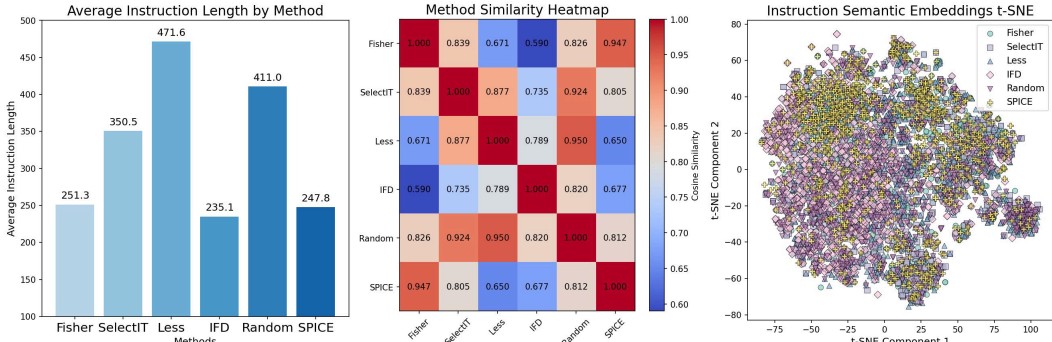

Figure 8: (a) Average instruction length of selected subset data by various methods. (b) Method similarity heatmap of instruction embeddings. (c) Instruction semantic embeddings t-SNE.

and memory overhead, going against SPICE's design goal of being a lightweight selection mechanism.

## H.5 STATISTICS OF SELECTED DATA

**Statistical Analysis.** In Section 5.4, we conducted statistical analysis on the selected 10% of data, first calculating the average instruction length of the data, and then random sample 10% of selected subset, to evaluate the semantic properties of the selected subsets, we embedding samples using all-MiniLM embeddings and visualize the embeddings with t-SNE and similarity heatmap.

As shown in Figure 8 (a), We can see that information based SPICE and Fisher tend to choose data with lower length, while Less selects data with the longest length; In (b) and (c) , the data we selected also have a relatively high similarity with Fisher, but through the trade-off of gradient conflict, we can select the data with better performance among them. In addition, we found that out of over 90000 data, only over 40000 data were selected under all data selection methods, and nearly half of the data points were not selected. Among them, math problems were selected the most.

| Baseline | Jaccard | Overlap@Ours | Overlap@Base |
|---|---|---|---|
| Fisher | 0.47 | 0.64 | 0.64 |
| Random | 0.05 | 0.10 | 0.09 |
| IFD | 0.02 | 0.03 | 0.03 |
| SelectIT | 0.08 | 0.15 | 0.15 |
| LESS | 0.01 | 0.01 | 0.02 |

Table 12: Pairwise overlap between baselines and our selection: Jaccard, Overlap@Ours, and Overlap@Base. Among them, Fisher, as a non-conflict-aware information-based method, has the highest overlap with SPICE's data.

**Overlap Analysis.** We summarize pairwise agreement with three set-overlap scores: Jaccard, Overlap@Ours, and Overlap@Base. Table 12 shows that Fisher exhibits the strongest agreement with our selector (Jaccard = 0.47; Overlap@Ours = 0.64; Overlap@Base = 0.64), indicating that both methods consistently capture a shared core of high-information examples. By contrast,

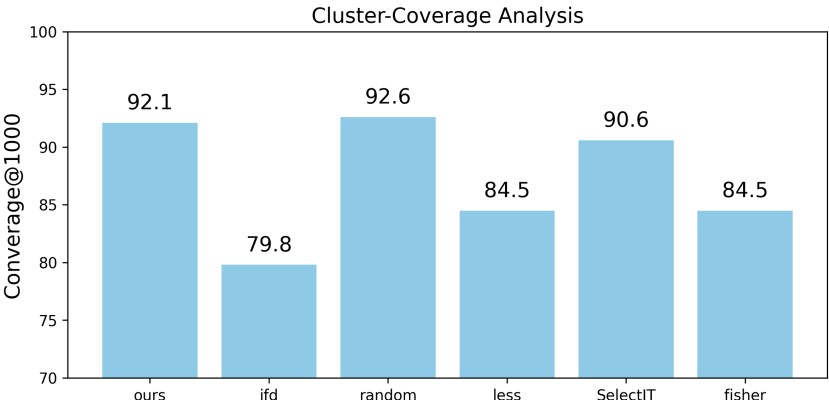

Figure 9: Brief cluster-hit histogram over $K=1000$ clusters; Coverage@1000 denotes the fraction of clusters with at least one selected item (higher indicates broader semantic coverage).

Random (Jaccard = 0.05; Overlap@Ours = 0.10; Overlap@Base = 0.09), IFD (0.02/0.03/0.03), SelectIT (0.08/0.15/0.15), and LESS (0.01/0.01/0.02) show substantially lower overlap with our picks, suggesting that their selection criteria emphasize different regions of the data space. This pattern supports the region that **our method preserves the high-information "core" while differing on more marginal or contentious items**. Building on this observation, we next turn to targeted case studies contrasting items jointly selected by non-conflict-aware information-based selection and ours.

**Cluster–Coverage Analysis.**  To verify that our selected subset spans the major semantic modes in the pool, we use the full data into $K=1000$ clusters (built once on instruction embeddings). We then map the already selected items onto these clusters and report a single coverage statistic, *Coverage@1000*, together with a brief cluster-hit histogram for context. Broad coverage (high Coverage@1000 with non-collapsed hits) indicates that our method surfaces high-quality examples from many data modes rather than concentrating on a few niches. This descriptive evidence helps explain why a 10% budget can outperform using all data: it preserves representative, informative items across clusters while avoiding redundant or contentious samples.

As shown in Figure 9, under the Coverage@1000 metric all methods cover over 80% of clusters. *Random* reaches 92.6, while *ours* is close behind at 92.1. Even with a 1,000-way partition, our subset still hits 92.1% of clusters, indicating broad semantic dispersion, i.e., **we consistently select at least one high-quality item per cluster**. This result helps explain how a $\sim$10% budget can match or surpass full-data training by capturing representative, informative examples while avoiding redundancy.

**Case Studies.**  As shown above, **we preserve the Fisher's informational core while removing boundary samples whose gradients strongly oppose the set's mean gradient direction, thereby mitigating marginal-gain decay and leading to a larger cumulative information**. Accordingly, we will analyze the high-conflict cases that were excluded and high-information cases that were overlapped; Table 13 lists 6 such cases.

### H.6   EXPANDED RESULTS AT 1% $\sim$30% DATA BUDGETS

In the main experiment (Section 5.2), we have shown that our method can outperform the full data on Qwen2-7B and LLaMA2-7B with 10% data. To assess robustness of our main conclusions under different data budgets, we replicate the full training-and-selection pipeline while varying only the subset fraction of the training pool. Concretely, we fix all model, optimization, and hyperparameters to the main-experiment (10%) values and run other three budget levels: 1%, 5% and $\sim$30% (SPICE+) compared to 100% (full data), Fisher (non-conflict-aware information-based method), DPP and Null (No fine-tunning). The results are shown in Table 14.

As shown in Table 14, across both base models the performance increases monotonically as the selection budget grows from 5% to $\sim$30%, with larger gains from 5%$\rightarrow$10% and smaller gains from

| Type | Instruction | Response |
|------|-------------|----------|
| E | Given the following list of words, come up with a sentence using as many of them as possible. Rainbow, Song, Cat, Inception | The rainbow appeared after the storm, and the cat curled up next to me as we listened to a beautiful song from the movie Inception. |
| E | What is a linked list? | A linked list is a data structure in which each element contains data as well as a link to the next element in the list. Linked lists provide a dynamic allocation of memory which can be useful for cer... |
| E | Select all rows in a pandas dataframe where at least one value exists in the column named 'name'. | selected_rows = df [df.name.notnull()] |
| S | (4 + 7) / 2 * 8 - 3= | The answer to this equation is 41. |
| S | Create a word that best describes the following definition. A person who is loyal despite difficulties | Resilient. |
| S | capitalize all the words in a string input: hello world | Hello World. |

Table 13: Case study of selections between non-conflict-aware information-based selection and SPICE: 3 excluded (E) and 3 shared (S) cases on default settings.

10%→∼30%. For example, on Qwen2-7B (SPICE) the average score rises from 56.5→58.0→58.1, where the 10%→∼30% gain is clearly smaller than the 5%→10% gain. LLaMA2-7B shows the same trend.

Compared with Full Data (100%), our method at 10% and ∼30% outperforms the full-data baseline on both models; at 1% and 5%, it is close to full data on Qwen2-7B but lower on LLaMA2-7B. **This indicates that even with tiny data, the diminishing-return benefits of information-based selection can outperform budget expansion.** Moreover, relative to Fisher, an information-based greedy selection that does not consider conflicts, our approach remains stronger even at the 1% and 5% budget, suggesting that conflicting samples lower overall gains. At the task level, our method delivers notably larger improvements on IFEval, indicating stronger benefits for instruction-following, while reasoning and knowledge benchmarks improve more steadily with the budget. Overall, the adaptive-stopping method (SPICE+) attains the best performance.

## H.7 COMPUTATIONAL COMPLEXITY

Standard Fisher-based greedy selection scales as $\mathcal{O}(k|D|d)$ due to repeated log-det evaluations, which is prohibitive for modern LLMs. In contrast, SPICE leverages (i) AdaFisher (Deb et al., 2025) to reduce the Fisher update cost from $\mathcal{O}(d^2)$ to $\mathcal{O}(d)$, (ii) a simple inner-product based conflict penalty that also costs $\mathcal{O}(d)$ per sample, and (iii) proxy models for gradient computation. Consequently, **the overall complexity is reduced to** $\mathcal{O}(k|D|d)$, which shown in Section 1. This linear-in-d complexity makes conflict-aware data selection feasible at billion-parameter scale (10B+), highlighting its scalability advantage.

Specifically, the detail pipeline is as followed:

**Step 1: Single-pass gradient computation** Each example $x \in D$ is used *once* to compute its per-sample gradient at a fixed proxy checkpoint. These gradients $g_x$ are *cached* and reused for all subsequent greedy steps.

$$\text{Gradient compute: } O(|D|d), \qquad \text{Memory: } O(md)$$

where $d$ is the proxy model dimension and $m$ is the candidate-pool size.

**Step 2: In-pool scoring** For each candidate pool $C$ with $|C| = m$:

- Fisher marginal gain (diagonal AdaFisher): $O(d)$ per example;

| Budget | Method | GSM8K | BBH | MMLU | ARC | T-QA | IFEval | H-Eval | MBPP | Avg |
|---|---|---|---|---|---|---|---|---|---|---|
| | | | | *Qwen2-7B* | | | | | | |
| 0% | Null | 77.8 | 60.3 | 64.4 | 48.3 | 54.3 | 25.8 | 43.9 | 53.8 | 53.6 |
| 1% | DPP | 77.9 | 60.4 | 64.3 | 48.4 | 54.2 | 26.0 | 44.8 | 53.8 | 53.5 |
| 10% | DPP | 86.5 | 61.0 | 66.0 | 51.0 | 55.0 | 35.4 | 45.0 | 55.7 | 57.0 |
| 1% | Fisher | 77.8 | 60.4 | 64.2 | 48.5 | 54.1 | 26.3 | 45.1 | 54.0 | 53.8 |
| 5% | Fisher | 83.4 | 60.5 | 65.2 | 50.1 | 54.5 | 30.6 | 43.7 | 54.0 | 55.3 |
| 10% | Fisher | 86.5 | 60.8 | 65.2 | 50.3 | 55.0 | 30.6 | 44.5 | 54.6 | 55.9 |
| ∼30% | Fisher | 86.5 | 61.1 | 67.0 | 50.8 | 55.2 | 33.8 | 45.2 | 56.2 | 57.0 |
| 1% | SPICE | 78.1 | 60.3 | 64.8 | 49.2 | 54.1 | 26.3 | 46.9 | 54.0 | 54.2 |
| 5% | SPICE | 83.5 | 60.4 | 64.8 | 50.5 | 55.2 | 35.5 | 46.7 | 55.0 | 56.5 |
| 10% | SPICE | 86.7 | 61.0 | 67.1 | 51.8 | **55.5** | 38.6 | 47.1 | 56.2 | 58.0 |
| ∼30% | SPICE+ | **86.7** | **61.3** | **67.3** | **52.0** | 55.5 | **38.6** | **47.5** | **56.2** | **58.1** |
| 100% | Full Data | 84.2 | 61.3 | 65.7 | 50.5 | 54.8 | 33.5 | 45.7 | 55.2 | 56.4 |
| | | | | *LLaMA2-7B* | | | | | | |
| 0% | Null | 12.7 | 39.9 | 39.9 | 43.1 | 38.8 | 18.8 | 14.0 | 23.0 | 28.8 |
| 1% | DPP | 12.7 | 40.0 | 40.0 | 44.0 | 38.9 | 19.1 | 14.5 | 23.0 | 28.9 |
| 10% | DPP | 13.6 | 40.2 | 41.0 | 47.0 | 40.0 | 22.2 | 15.9 | 23.5 | 30.4 |
| 1% | Fisher | 13.0 | 40.0 | 40.3 | 43.5 | 38.8 | 19.0 | 14.4 | 22.8 | 29.0 |
| 5% | Fisher | 13.4 | 40.1 | 41.0 | 46.2 | 39.3 | 21.8 | 15.1 | 23.0 | 30.0 |
| 10% | Fisher | 13.6 | 40.5 | 41.5 | 46.7 | 39.3 | 22.0 | 15.2 | 23.2 | 30.3 |
| ∼30% | Fisher | 13.7 | 41.4 | 41.9 | 47.1 | 40.1 | 22.5 | 16.7 | 24.3 | 31.0 |
| 1% | SPICE | 12.7 | 40.1 | 40.1 | 45.9 | 38.9 | 20.0 | 14.6 | 22.4 | 29.3 |
| 5% | SPICE | 12.8 | 40.5 | 40.5 | 46.0 | 40.5 | 21.0 | 16.0 | 24.0 | 30.2 |
| 10% | SPICE | 13.8 | 40.8 | 41.9 | **47.7** | 40.3 | 22.6 | 16.7 | 24.6 | 31.1 |
| ∼30% | SPICE+ | **13.8** | **41.5** | **42.1** | 47.3 | 40.3 | **22.6** | **16.9** | 24.6 | **31.2** |
| 100% | Full Data | 13.6 | 41.3 | 40.8 | 46.1 | **43.5** | 19.4 | 16.5 | **25.2** | 30.8 |

Table 14: SPICE per-benchmark scores at 1%, 5%, 10%, and ∼30% budgets compared to 0% (Null), 100% (full data), Fisher (non-conflict-aware information-based method) and DPP. Among them, **T-QA** represents TruthfulQA, **H-eval** represents HumanEval dataset, and ∼30% means that SPICE+ is used (26.1% for Qwen2-7B and 29.8% for LLaMA2-7B).

- Conflict score (cosine w.r.t. mean gradient): $O(d)$ per example.

Thus, one scoring pass costs

$$O(md).$$

**Step 3: In-pool greedy selection** For each pool, selecting $k_{\text{pool}}$ examples requires rescoring the remaining candidates:

$$O(k_{\text{pool}}\, m\, d)\,.$$

Summing across pools with $\sum k_{\text{pool}} = k$ gives

$$O(k\, |D|\, d).$$

**Step 4: Proxy model updates (interval $T$)** SPICE updates the proxy model once every $T$ pools, which adds

$$O(kd),$$

negligible relative to the selection cost.

**Step 5: Overall time and space complexity**

$$T_{\text{total}} = O(|D|d) + O(k|D|d) + O(kd) = O(k\, |D|\, d).$$

Peak memory usage is

$$O(md),$$

since we only store gradients for one pool at a time.

**Summary** SPICE is a *single-pass, streaming* procedure: (i) each example in $D$ produces exactly one per-sample gradient; (ii) all subsequent selection steps reuse cached gradients; (iii) no full-dataset recomputation of gradients is performed; (iv) complexity scales linearly with dataset size and model dimension.

## H.8 TIME COST

All experiments run on the same 8×H20 GPU node with a shared LoRA SFT setup across methods (3 epochs, same batch size, max length, optimizer, LR schedule, and LoRA hyper-parameters). Under a fixed 10% data budget, fine-tuning cost is therefore identical across methods, and the main difference comes from selection time.

Table 15 reports selection time on this 8-GPU node together with average performance. SPICE needs only 2:56 of selection time, and its total time (selection + 10% LoRA SFT) is lower than full-data LoRA, while achieving higher performance on both Qwen2-7B and LLaMA2-7B.

| Method | Selection Time Cost (hour:min) | Selection Computation Complexity | Performance (Qwen2-7B) | Performance (LLaMA2-7B) |
|---|---|---|---|---|
| **Full** | 00:00 | $O(0)$ | 56.4 | 30.8 |
| **Random** | 00:00 | $O(k)$ | 55.5 | 30.1 |
| **SPICE** | 02:56 | $O(k\|D\|d)$ | **58.0** | **31.1** |
| **IFD** | 04:32 | ~ | 55.4 | 30.0 |
| **LESS** | 16:22 | $O(Nm\|D\|d)$ | 55.0 | 30.3 |
| **Fisher** | 17:01 | ~ | 55.2 | 30.2 |
| **SelectIT** | 25:19 | ~ | 55.8 | 30.1 |
| **DPP** | 23:32 | $O(NMD + ND)$ | 56.3 | 30.5 |
| **TSDS** | 00:05 | $O(ML\log N)$ | 55.4 | 29.9 |
| **LEAD** | 12:02 | ~ | 56.5 | 31.0 |

Table 15: Time cost and computation complexity.

# I   LARGE LANGUAGE MODELS USAGE

Did you use Large Language Models (LLMs) in paper writing? Yes, LLMs were used solely for polishing writing and formatting of equations. No content, data analysis, or interpretation was generated or influenced by LLMs. All research results and conclusions were independently developed by ours.

