# OpenReview forum: "SPICE: Submodular Penalized Information–Conflict Selection for Efficient Large Language Model Training"
_ICLR.cc/2026/Conference — ICLR 2026 Poster_

### Official Review · Reviewer_KYWw · 2025-10-26

**Soundness:** 3
**Presentation:** 3
**Contribution:** 4
**Rating:** 8
**Confidence:** 3

**Summary:**

This paper addresses the problem of selecting a small subset of instruction-tuning data for large language models in order to fine-tune efficiently. The authors observe that while maximizing the log-determinant of the empirical Fisher information matrix yields a submodular objective, in practice marginal gains collapse quickly due to gradient misalignment among samples. They formalize this by decomposing the marginal information gain into a base term and an interaction term, and conclude that controlling gradient conflict is key to sustaining information gain. Based on this, they propose SPICE: a selection algorithm that (1) uses a scoring function that subtracts a conflict penalty from the marginal information gain, (2) optionally stops early once the marginal gain falls below a threshold, and (3) uses a proxy (smaller) model to compute the gradients efficiently. They empirically show that on multiple benchmarks, using only ~10% of data, SPICE matches or exceeds full-data fine-tuning and outperforms several baselines, while reducing computation cost.

**Strengths:**

The method addresses both effectiveness (maintaining or improving performance with fewer training samples) and efficiency (using proxy model selection & early stopping) — a nice combination.

Empirical results are broad (multiple benchmarks, models, tasks) and show impressive savings (≈10% data) with no performance loss and even gains in some cases.

The algorithm is extensible: the idea of “penalize conflict” could be applied in other data-selection or multi-task contexts.

**Weaknesses:**

The current scope of experiments is limited to ~7 B-parameter models and instruction-tuning; extension to larger models (>30 B), multimodal tasks, or RLHF settings remains to be seen.

The proxy-to-target model transfer is shown only within same architecture family; cross-architecture transfer (e.g., completely different model family) may degrade and is less explored.

The cost comparison, while present, could be strengthened with more granular breakdowns (selection cost vs fine-tune cost) across all baselines under identical hardware settings.

The penalty on “conflict” implicitly biases toward samples aligned with current gradient direction—there is a risk that samples with contradictory but important signals might be under-selected; more analysis of diversity vs conflict trade-offs would help.

**Questions:**

How sensitive is SPICE to the choice of proxy model? If the proxy model differs in architecture or domain from the fine-tune target, how does performance vary?

In domains with heterogeneous instruction types (e.g., chat, coding, planning) where gradient directions may naturally differ, how does the conflict penalty trade off between “reducing harm” vs “reducing diversity”? Have you analysed domain-coverage of the selected subset?

Could you provide more detailed hardware/GPU-hour breakdowns (selection + fine-tune) for each baseline method (e.g., LESS, SelectIT, FisherSFT) under identical hardware, to strengthen the cost-efficiency claim?

Have you tested SPICE on larger models (>30 B) or other modalities (vision+language) or RLHF settings? If not yet, what do you foresee as the main challenge in scaling?

---

> ### Author Response · Authors · 2025-11-20
> **Response to Reviewer KYWw**
>
> **`Question1&4 Weakness1&2:` Sensitivity of the choice of proxy model, Cross-architecture transfer and Model scale**
>
> **`Respones1:`** We appreciate the reviewer’s question about how SPICE depends on the choice of proxy model. We investigate this in two ways:
>
> 1. Proxy Model Sensitivity:
> In Appendix H.5, we analyze different proxies for selecting data for a fixed target model (Qwen2-7B). We vary the proxy within the same architecture family (Qwen2-0.5B, 1.5B, 7B) and also use a cross-family proxy (LLaMA2-7B). When using proxies from the same family, the selected subsets show high overlap, and downstream LoRA SFT performance differs by less than ~0.3 points, indicating that SPICE is robust to proxy size as long as the proxy and target share the same architecture. In contrast, using LLaMA2-7B as a proxy for Qwen2-7B results in a consistent but modest drop, highlighting the challenge of cross-architecture gradient transfer.
>
> 2. Scaling to Larger Models:
> We extend our experiments to 70B+ target models while using small same-family proxies (0.5B/7B). Even when fine-tuning 70B-scale models, SPICE subsets selected by small proxies match or outperform full-data LoRA, showing that very large proxies are not necessary in practice. Using models from 30B–70B as proxies would require significant computational and memory overhead, going against SPICE's design goal of being a lightweight selection mechanism.
>
> We will (1) add the sensitivity to proxy model choice and its empirical results in the revised main text, (2) explicitly highlight that SPICE-selected 10% subsets match or outperform full-data LoRA even for 30B–70B models, and (3) clarify that very large models as proxies are not required for SPICE to scale to large base LMs.
>
>
> | SFT Model  | SPICE (Proxy Model- Qwen2-0.5B) | SPICE (Proxy Model- Qwen2-7B) | SPICE (Proxy Model- LLaMA-7B) | Null | Full |
> |------------|---------------------------------|-------------------------------|--------------------------------|------|------|
> | Qwen2-0.5B | 31.1                            | 30.8                          | 29.9                           | 29.8 | 30.5 |
> | Qwen2-7B   | 67.0                            | 67.2                          | 66.1                           | 62.0 | 65.2 |
> | Qwen2-72B  | 77.9                            | 78.0                          | 77.4                           | 77.2 | 77.5 |
> | LLaMA2-7B  | 23.6                            | 23.4                          | 24.1                           | 22.2 | 23.6 |
> | LLaMA2-70B | 51.4                            | 51.4                          | 50.9                           | 50.9 | 51.1 |

---

> ### Author Response · Authors · 2025-11-20
> **Response to Reviewer KYWw [Part 2]**
>
> **`Question2 & Weakness4:` Diversity analysis**
>
> **`Response2:`** We thank the reviewer for the thoughtful question regarding whether the conflict penalty might suppress “disagreeing but useful” examples in heterogeneous domains. In SPICE, conflict is implemented as a **soft** regularizer on top of the Fisher marginal gain:
> $$
> \mathrm{score}(x\mid S)=\Delta_x(S)-\lambda\cdot \mathrm{Conflict}(x\mid S),
> $$
> so examples with high information gain but partially misaligned gradients are still selected. Our λ-ablation (Sec. 5.4) shows that moderate λ improves over λ=0, while large λ provides no further benefit—indicating that SPICE filters **harmful conflicts** rather than removing beneficial diversity.
>
> To directly evaluate whether semantic or domain diversity is reduced, we report (i) **domain coverage** across *code*, *math/reasoning*, and *general* instructions, and (ii) two diversity metrics proposed in Yang et al. (ACL 2025): **LDD** and **NovelSum**. Results are shown below.
>
> ------
>
>
> | Method (10%) | LDD  | NovelSum | Domain Coverage ( Code / Math Reasoning / General) |
> |--------------|------|----------|----------------------------------------------------|
> | SPICE        | 22.0 | 41.3     | 10% / 8% / 9%                                      |
> | SPICE+       | 28.9 | 40.2     | 11% / 7% / 9%                                      |
> | Random       | -9.5 | 30.3     | 5% / 10% / 10%                                     |
> | IFD          | -8.4 | 34.3     | 3% / 4% / 14%                                      |
> | Fisher       | 19.4 | 40.3     | 12% / 2% / 9%                                      |
> | LESS         | 4.9  | 38.7     | 1% / 12% / 11%                                     |
> | SelectIT     | 17.6 | 39.0     | 8% / 13% / 9%                                      |
> | TSDS         | -1.8 | 34.8     | 4% / 5% / 10%                                      |
> | DPP          | 31.1 | 42.5     | 7% / 7% / 9%                                       |
> | LEAD         | 9.8  | 37.5     | 2% / 6% / 11%                                      |
>
>
>
> SPICE and SPICE+ achieve **high diversity** (LDD 22–29; NovelSum 40–41) comparable to the diversity-oriented DPP baseline, and clearly higher than Random, LESS, and SelectIT. Domain fractions (e.g., 10%/8%/9%) closely match the full corpus distribution, showing that SPICE maintains balanced coverage across heterogeneous instruction types.
>
> [1] Measuring Data Diversity for Instruction Tuning: A Systematic Analysis and A Reliable Metric (Yang et al., ACL 2025)

---

> ### Author Response · Authors · 2025-11-20
> **Response to Reviewer KYWw [Part 3]**
>
> **`Question3 & Weakness3:` Cost comparison and hardware**
>
> **`Response3:`** We thank the reviewer for this suggestion. All experiments run on the same 8×H20 GPU node with a **shared LoRA SFT setup** across methods (3 epochs, same batch size, max length, optimizer, LR schedule, and LoRA hyperparameters; Sec. 4.3 / App. H.3). Under a fixed 10% data budget, fine-tuning cost is therefore identical across methods, and the main difference comes from **selection time**.
>
> The table below reports selection time on this 8-GPU node together with average performance. SPICE needs only 2:56 of selection time, and its total time (selection + 10% LoRA SFT) is **lower than full-data LoRA**, while achieving **higher** performance on both Qwen2-7B and LLaMA2-7B.
>
> | Method   | Selection Time (hour:min) | Performance (Qwen2-7B) | Performance (LLaMA2-7B) |
> |----------|---------------------------|-------------------------|--------------------------|
> | Full     | 00:00                     | 56.4                    | 30.8                     |
> | Random   | 00:00                     | 55.5                    | 30.1                     |
> | SPICE    | 02:56                     | 58.0                    | 31.1                     |
> | IFD      | 04:32                     | 55.4                    | 30.0                     |
> | LESS     | 16:22                     | 55.0                    | 30.3                     |
> | Fisher   | 17:01                     | 55.2                    | 30.2                     |
> | SelectIT | 25:19                     | 55.8                    | 30.1                     |
> | DPP      | 23:32                     | 56.3                    | 30.5                     |
> | TSDS     | 00:05                     | 55.4                    | 29.9                     |
> | LEAD     | 12:02                     | 56.5                    | 31.0                     |
>
>
>
>
>
>
>
>
>
> **`Question4:` Future work**
>
> **`Response4:`** We thank the reviewer for highlighting the potential of extending SPICE to more tasks. SPICE builds on the Fisher matrix as an interpretable, task-aware metric, together with efficient approximations like AdaFisher, which makes the approach naturally scalable.
>
> Looking ahead, we are actively studying Fisher-based selection in the *pre-training* regime (integrating conflict-aware sampling into the training loop), as well as RLHF settings where Fisher geometry can be used in a GRPO-style algorithm to modulate preference data. For multimodal tasks (e.g., vision + language), the main challenge is to define Fisher and conflict over multimodal representations, and we view this as a promising direction building on the current SPICE framework.

---

### Official Review · Reviewer_Xhgm · 2025-10-30

**Soundness:** 3
**Presentation:** 2
**Contribution:** 3
**Rating:** 6
**Confidence:** 3

**Summary:**

This paper proposes a submodular framework for data-efficient language model fine-tuning, introducing a conflict-aware selection mechanism that balances information gain and gradient disagreement. The method improves greedy subset selection efficiency and achieves competitive results.

**Strengths:**

1. Novel insight into the fast decay of marginal contribution to enhance greedy submodular optimization.
2. Comprehensive experiments and ablations supporting the method’s effectiveness.
3. Decent performance on various benchmarks.

**Weaknesses:**

1. High computational cost due to gradient retrieval for each selection step.
2. Possible misalignment between the theoretical motivation and empirical design; the overall selection pipeline remains somewhat unclear (see questions).
3. Limited baseline comparisons (see questions).

**Questions:**

1. *Theorem 1 (rows 167–169):* Why do large perturbations lead to faster decay? Shouldn’t it be the difference between successive perturbations, not the absolute magnitude of a perturbation, that drives faster decay?
2. *Definition 4:* Corollary 1 penalizes both similar and opposite gradients via squared inner products, while Definition 4 only penalizes opposite ones. Why are similar gradients (redundancy) ignored, given that the theory penalizes both?
3. *Pipeline clarity:*
    - Section 4.3 (row 346): When stating “at each iteration, we select one sample using our conflict-aware greedy algorithm,” does ‘one sample’ refer to a single example or a mini-batch of k samples? If it refers to a single example, does the model get updated after each selection (get updated after seeing a new example) when T=1?
    - Section 5 (row 372): How is the 120-sample candidate pool formed? Is it randomly drawn from D with size k×T?
    - Is the proxy model updated after each cycle?
4. *Baselines and related work:*
    - If the proxy model is periodically updated and selection occurs within a randomly sampled “candidate pool”, the setup seems closer to online batch selection, making comparisons to FisherSFT, LESS, or IFD, a non-periodic selection mechanism, potentially unfair. It remains unclear whether SPICE’s performance gains stem from the periodic schedule or the proposed selection mechanism.
    - Representation-based selection methods [1] and other recent instruction-tuning data selection works [2-4] are not discussed.
5. *Complexity analysis:* Could the authors provide an explicit asymptotic analysis of time complexity? Algorithm 1 appears to require gradient computation over the entire dataset D for each selection, which seems computationally expensive.

[1] Ivison, H., Zhang, M., Brahman, F., Koh, P. W., & Dasigi, P. (2025). *Large-Scale Data Selection for Instruction Tuning*. arXiv preprint arXiv:2503.01807.

[2] Liu, Z., Karbasi, A., & Rekatsinas, T. (2024). *TSDS: Data Selection for Task-Specific Model Finetuning*. In *The Thirty-eighth Annual Conference on Neural Information Processing Systems (NeurIPS 2024)*.

[3] Wang, J., Lin, X., Qiao, R., Koh, P. W., Foo, C.-S., & Low, B. K. H. (2025). *NICE Data Selection for Instruction Tuning in LLMs with Non-differentiable Evaluation Metric*. In *Forty-second International Conference on Machine Learning (ICML 2025)*.

[4] Chen, Y., Li, Y., Hu, K., Ma, Z., Ye, H., & Chen, K. (2025). *MIG: Automatic Data Selection for Instruction Tuning by Maximizing Information Gain in Semantic Space*. In *Findings of the Association for Computational Linguistics: ACL 2025*.

---

> ### Author Response · Authors · 2025-11-20
> **Response to Reviewer Xhgm**
>
> **`Question5 & Weakness1:` Computational cost and Complexity analysis**
>
>
>
>
>
> **`Response1:`** We thank the reviewer for the thoughtful question. Algorithm 1 is a conceptual offline description; the actual implementation of SPICE is a **streaming algorithm** that performs exactly **one gradient computation per example** and never recomputes gradients over the full dataset.
>
> While Appendix H.8 already includes a brief complexity analysis, below we provide a more detailed introduction, and we will expand Appendix H.8 accordingly in the revised version. We hope the following content will help better understand the cost and practicality of SPICE:
>
> **Step 1: Single-pass gradient computation**
>
> > Each example $x \in D$ is used once to compute a per-sample gradient at a fixed proxy checkpoint.
>  These gradients $g_x$ are **cached** and reused for all later greedy steps.
> $$
> \text{Gradient compute: } O(|D|d), \quad
> \text{Memory: } O(md)
> $$where $d$ is the proxy model dimension and $m$ is the candidate pool size.
>
> **Step 2: In-pool Scoring**
>
> > For each candidate pool $C$ with $|C| = m$:
> >
> > - Fisher marginal gain (diagonal AdaFisher): $O(d)$ per example
> > - Conflict score (cosine with mean gradient): $O(d)$ per example
> >
> > Thus, one scoring pass costs:
> > $$
> > O(md)
> > $$
>
> **Step 3: In-pool Greedy Selection**
>
> > For each pool, selecting $k_{\text{pool}}$ examples requires rescoring the remaining candidates:
> $$
> O(k_{\text{pool}} m d)
> $$
> Summing across pools with $\sum k_{\text{pool}} = k$:
> $$
> O(k |D| d)
> $$
>
> **Step 4: Proxy Model Updates (Step Interval $T$)**
>
> > SPICE updates the proxy model once every $T$ pools, adding:
> $$
> O(kd)
> $$which is negligible relative to the selection cost.
>
> **Step 5: Overall Time and Space Complexity**
>
> > $$
> T_{\text{total}}
> = O(|D|d) + O(k|D|d) + O(kd)
> = O(k |D| d)
> $$
> >
> > Peak memory usage is:
> $$
> O(md)
> $$ since we only store gradients for one pool at a time.
>
>
> SPICE is a **single-pass, streaming** procedure:
>
> - each example in $D$ produces exactly **one** per-sample gradient;
> - all subsequent selection steps use cached gradients;
> - **no full-dataset recomputation** of gradients is performed;
> - complexity scales linearly in dataset size and model dimension.
>
> This ensures that SPICE remains computationally practical even for large datasets and proxy models.

---

> ### Author Response · Authors · 2025-11-20
> **Response to Reviewer Xhgm [Part 2] (revised: fix typo)**
>
> **`Question3 & Weakness2:` Clarity for SPICE pipeline**
>
> **`Response2:`** We sincerely appreciate the thoughtful questions about the SPICE pipeline. Following the detailed explanation provided in **`Response1`**, we now address each of the reviewer’s comments:
>
> **Question 3.(1) Section 4.3 (row 346): The meaning of "one sample":**
>
> > In Section 4.3, the phrase “at each iteration, we select one sample” refers to a **greedy step within a candidate pool**, not an online per-example update.
> At each greedy iteration we pick one example from the pool and add it to the
> accumulating subset. The proxy model is **not** updated after each selected sample; it is updated only once every \(T\) pools according to the step-interval schedule.We will clarify this description in the revision to avoid ambiguity.
>
> **Question 3.(2) Section 5 (row 372): How is the 120-sample candidate pool formed? Is it randomly drawn from D with size $k\times T$?**
>
> > Each candidate pool (we use \(m = 120\) by default) is formed by uniformly sampling \(m\) examples from \(D\) with a fixed random seed for reproducibility. The term \(k $\times$ T\) refers to the accumulated number of selected examples used for proxy updates, not the pool size.We also verified that changing the sampling seed or batch size produces only small variations in the selected subset (Jaccard overlap ≈ 0.7–0.8) while the downstream performance remains essentially unchanged (≈ 67), confirming that the choice of pool sampling has minimal impact on SPICE.
>
>
> **Question 3.(3) Is the proxy model updated after each cycle?:**
>
> >  No. The proxy model is not updated after each cycle (i.e., after each candidate pool). It is updated only every $T$ candidate pools, according to the step-interval schedule. To further clarify, in Section 5.4 (row 429) and Appendix H.5 we provide an ablation study on the step-interval used for proxy updates. The results show that the performance remains stable across the update intervals.
>
>
>
>
> **`Question1:` Theorem 1: why do large perturbations lead to faster decay? (Question 1)**
>
> **`Response3:`** We thank the reviewer for pointing out the wording around rows 167–169. At the level of instantaneous (single-step) decay, the decrease in marginal gain between two nested sets is indeed controlled by the difference between successive perturbations, as in Eq. (5): $\Delta_x(A) - \Delta_x(B) = \varepsilon_x(A) - \varepsilon_x(B)$,
> rather than by the absolute value $|\varepsilon_x(S)|$ at a single set.
>
> What we ultimately analyze, however, is the global behavior of greedy selection via the total curvature (Eq. (7)):
> $$c \triangleq 1 - \min_{x \in D} \Delta_x(D \setminus {x}) / \Delta_x(\emptyset) \in [0,1],$$ which yields the standard curvature-based guarantee (Eq. (8)):
> $$F(S_{\text{greedy}}) \ge \frac{1 - e^{-c}}{c} \cdot F(S^*).$$ Here, $c$ measures the cumulative normalized decay of marginal gains.
>
> Our $\varepsilon$-decomposition links this cumulative decay to perturbations. In the Fisher setting, we write $\Delta_x(S) = \text{base}x + \varepsilon_x(S)$, so along any chain $\varnothing = S_0 \subset \cdots \subset S_T = D \setminus {x}$:
>
> $$\Delta_x(\varnothing) - \Delta_x(D \setminus {x}) = \sum_{t=1}^T (\Delta_x(S_{t-1}) - \Delta_x(S_t)) = -\varepsilon_x(D \setminus {x}) = |\varepsilon_x(D \setminus {x})|.$$
> Thus the total curvature can be rewritten as
> $$c = - \min_x \varepsilon_x(D \setminus {x}) / \text{base}_x \le \max_x |\varepsilon_x(D \setminus {x})| / \text{base}_x,$$ showing that larger perturbations on large sets imply larger cumulative decay and hence larger curvature (weaker greedy guarantees).
>
> We thank the reviewer for the helpful clarification. To reflect this more accurately, we will revise the sentence around lines 167–169 to state that **$|\varepsilon_x(S)|$ characterizes the cumulative decay of the marginal gain** from its base value and controls the total curvature $c$, rather than claiming it directly controls the instantaneous decay rate in Eq. (5).

---

> ### Author Response · Authors · 2025-11-20
> **Response to Reviewer Xhgm [Part 3]**
>
> **`Question2`: Definition 4 and Corollary 1’s squared inner products (Question 2)**
>
> **`Response4:`** Thanks for raising this point. In Corollary 1, the squared inner products appear inside a worst-case curvature bound: they control how gradient interactions (both aligned and opposite) can cumulatively shrink marginal gains. In the algorithm, we deliberately separate these two effects instead of penalizing them twice.
>
> Redundancy from **similar** gradients is already handled by the Fisher term $\Delta_x(S)$: when many examples point in a similar direction, their marginal Fisher gains decay quickly (via the $\varepsilon_x(S)$ term), so the greedy Fisher objective naturally stops selecting highly redundant points. Definition 4 therefore uses the conflict term only to penalize **opposite** gradients—those that actively undo the progress made by the current subset—rather than re-penalizing redundancy that the Fisher log-det term already suppresses.
>
> To probe this design, we add an ablation where we fix the Fisher marginal term $\Delta_x(S)$, keep the 10% budget and SFT setup fixed, and only vary the conflict penalty:
>
> 1. $\lambda = 0$: pure Fisher-greedy (no conflict term)
> 2. $\lambda = 0.1$: one-sided SPICE conflict (penalize opposite only)
> 3. $\lambda = 0.1$: symmetric alignment penalty $|\mathrm{Align}(g_x,\bar g_S)|$ (penalize aligned & opposite)
> 4. $\lambda = 0.1$: symmetric inner-product penalty $|g_x^\top \bar g_S|$ (penalize aligned & opposite)
>
> | Variant ID | λ   | Conflict term                              | Penalizes          | Performance (Qwen2-7B) | NovelSum |
> |------------|-----|--------------------------------------------|--------------------|------------------------|----------|
> | (1)        | 0   | –                                          | none               | 65.3                   | 41.0     |
> | (2)        | 0.1 | $\max(0,\,-\mathrm{Align}(g_x,\bar{g}_S))$ | opposite only      | 67.0                   | 41.3     |
> | (3)        | 0.1 | $\|\mathrm{Align}(g_x, \bar{g}_S)\|$       | aligned & opposite | 65.9                   | 35.4     |
> | (4)        | 0.1 | $\|g_x^\top \bar{g}_S\|$                   | aligned & opposite | 66.2                   | 36.8     |
>
> Turning off the conflict term ($\lambda = 0$) clearly hurts performance compared to SPICE, showing that explicitly downweighting strongly opposing gradients is beneficial. The symmetric penalties that also penalize aligned gradients do not improve over SPICE and noticeably reduce NovelSum diversity, consistent with the view that redundancy among similar gradients is already controlled by the Fisher gains, and that the conflict term is most useful when it focuses on destructive interference. In the revision we will make this separation of roles explicit and include this ablation in the main text.
>
> [1] Measuring Data Diversity for Instruction Tuning: A Systematic Analysis and A Reliable Metric (Yang et al., ACL 2025)
>
> **`Question4-1:` Periodic schedule (Question 4.(1))**
>
> **`Response5:`** We appreciate the concern that periodically updating the proxy model and selecting within randomly sampled candidate pools might make SPICE closer to an “online batch selection” setup, potentially making comparisons to offline baselines.
>
> SPICE is online only with respect to the **proxy** model, not the **target** model. During selection, the proxy is updated every $T$ candidate pools using the already selected data, to better align its gradients with the evolving subset. However, for all methods (SPICE, Fisher, LESS, IFD, DPP, TSDS, LEAD, Random, Full), the **target model** is always trained in a standard offline fashion: we first select a 10% subset, then run the same LoRA SFT schedule on that subset. No method interleaves selection with target-model training.
>
>
> To separate the effect of the periodic schedule from the selection mechanism, we add an ablation over the step interval $T$. As shown below and in Section 5.4, removing proxy updates ($T=0$) only slightly reduces performance, and varying $T$ from 1 to 50 changes the average score by at most about 0.4–0.6 points:
>
>
> | Step Intervals | Performance (Qwen) |
> |----------------|--------------------|
> | 0 (no update)  | 66.5               |
> | 1              | 67.0               |
> | 5              | 67.0               |
> | 50             | 66.6               |
>
> Even with $T=0$ (no periodic proxy update), SPICE still **outperforms** all offline subset-selection baselines (Random, IFD, LESS, Fisher, SelectIT) under the same 10% budget, indicating that the main gains come from the conflict-aware Fisher criterion rather than the update schedule. In addition, we include TSDS and LEAD as online/iterative selection baselines; both also operate in an online fashion, yet SPICE **achieves higher average performance**. Taken together, these results show that SPICE’s **advantage is due to the proposed conflict-aware Fisher objective, not to a particular scheduling trick**.

---

> ### Author Response · Authors · 2025-11-20
> **Response to Reviewer Xhgm [Part 4]**
>
> **`Question4-2 & Weakness3:` Baselines and related work**
>
> **`Response6:`** We thank the reviewer for the insightful comments on baselines and on the connection between our periodic schedule and online batch selection.
>
> In the revised version, we add three additional baselines under this line of work:
>
> - TSDS, a recent task-specific data selection method that is conceptually closest to our setting;
> - LEAD , an online iterative data selection method for instruction tuning;
> - DPP, a determinantal point process on representation features as a diversity-oriented baseline (mentioned by Reviewer Y2FZ).
>
> All newly added methods are run under the same 10% data budget and training pipeline as SPICE and the existing baselines: for every method we first select a 10% subset according to its own criterion, and then fine-tune the target model with the same LoRA SFT settings in Sec. 4.3. This ensures that performance differences come from the selection criterion, not from different training schedules.
>
> As shown below, we report average performance over 8 benchmarks together with selection time and computation complexity. In this online-selection setting, SPICE achieves a **favorable cost–performance trade-off compared to the added baselines** and outperforms all three newly added baselines. These baselines are incorporated into Tables 1 and 6 (rows 380 and 1620) in the revised manuscript.
>
> | Method   | Selection Time Cost (hour:min) | Selection Computation Complexity | Performance (Qwen2-7B) | Performance (LLaMA2-7B) |
> |----------|--------------------------------|----------------------------------|------------------------|-------------------------|
> | Full     | 00:00                          | O(0)                             | 56.4                   | 30.8                    |
> | Random   | 00:00                          | O(k)                             | 55.5                   | 30.1                    |
> | SPICE    | 02:56                          | O(k \|D\| d)                     | 58.0                   | 31.1                    |
> | IFD      | 04:32                          | ~                                | 55.4                   | 30.0                    |
> | LESS     | 16:22                          | O(Nm \|D\| d)                    | 55.0                   | 30.3                    |
> | Fisher   | 17:01                          | ~                                | 55.2                   | 30.2                    |
> | SelectIT | 25:19                          | ~                                | 55.8                   | 30.1                    |
> | DPP      | 23:32                          | O(NMD + ND)                      | 56.3                   | 30.5                    |
> | TSDS     | 00:05                          | O(ML log N)                      | 55.4                   | 29.9                    |
> | LEAD     | 12:02                          | ~                                | 56.5                   | 31.0                    |

---

> > ### Comment · Reviewer_Xhgm · 2025-11-28
> >
> > Thank you for the additional experiments and clarifications, especially the new results in Response 4, which effectively address the potential gap between the theoretical setup and the algorithmic design. Most of my earlier concerns have also been resolved.
> >
> > However, I remain concerned that Question 1 is still not fully addressed. The revision to lines 167–169 in the current version can be confusing without the accompanying proofs in response 3 (and there appears to be a typo in the second-to-last equation, where a subscript after the “$\sum$” term seems to be missing). The new analysis relies on the introduction of total curvature, but this concept is introduced only after Theorem 1 in the paper. This suggests that the analysis added in the rebuttal may need a re-ordered presentation to improve coherence.
> >
> > The additional analysis in the rebuttal addresses my concern to some extent, but it also highlights that certain edits may be necessary to make the overall flow more consistent. The revised PDF still does not yet fully resolve this structural issue. Therefore, I will maintain my current score of weak accept for now.

---

> > > ### Author Response · Authors · 2025-11-28
> > > **Response to Reviewer Xhgm**
> > >
> > > Dear Reviewer Xhgm,
> > >
> > > Thank you very much for your careful follow-up. We appreciate the constructive clarification of your remaining concern. Based on your suggestion, we made structural revisions to ensure that the explanation of Question 1 is fully coherent without relying on material introduced later in the section and rebuttal-only derivations.
> > >
> > > **(A) Structural re-organization for coherence**
> > >
> > > To avoid the earlier logical jump, we have re-ordered the presentation such that the $\varepsilon$-decomposition, the chain argument (Eq. (6)), and the interpretation of $|\varepsilon_x (S)|$ as the cumulative decay of marginal gains now appear entirely before the introduction of total curvature in Section 2.3. This ensures that the answer to Question 1 is self-contained within Section 2.2, requires no forward reference to curvature, and does not rely on any rebuttal-only derivation. The theoretical flow is now:
> > >
> > > Theorem 1 → Eq.(5) → chain argument (Eq.(6)) → interpretation → (later) curvature extension.
> > >
> > > We believe this directly resolves the structural issue you highlighted.
> > >
> > > **(B) Focused edits**
> > >
> > > 1. **Clarification around row 167–169.**
> > > We rewrote the paragraph following Theorem 1 to give a short, explicit chain argument showing that $|\varepsilon_x (S)|=\Delta_x(\varnothing) - \Delta_x (S)$, i.e., $\varepsilon$ captures the cumulative decay of marginal gain. This explanation is now fully self-contained.
> > >
> > > 2. **Explicit connection to curvature (now in Section 2.3 only, row 203-205).**
> > > After introducing total curvature and Lemma 1, we added a concise sentence expressing the curvature numerator in terms of $\varepsilon_x (D \setminus {x})$. This makes the $\varepsilon \rightarrow$curvature link explicit, while keeping curvature separate from the explanation of Q1.
> > >
> > > 3. **Equation typo in `rebuttal-Response 3.`** We have corrected the missing subscript in the summation expression: $\sum{t=1}^T (\Delta_x(S_{t-1}) - \Delta_x(S_t))   \Rightarrow \sum_{t=1}^T (\Delta_x(S_{t-1}) - \Delta_x(S_t)) $
> > >
> > > These edits do not modify any theorem or claim, but they significantly improve the clarity and coherence of the presentation. All changes are already included in the revised manuscript and our summary of revision.
> > >
> > > We thank you again for your thoughtful feedback and for helping us improve the clarity of the paper.
> > >
> > > Sincerely,
> > >
> > > The Authors

---

### Official Review · Reviewer_rHBr · 2025-10-31

**Soundness:** 3
**Presentation:** 4
**Contribution:** 3
**Rating:** 6
**Confidence:** 2

**Summary:**

The paper studies selecting instruction-tuning datasets by proposing to avoid gradient conflicts. The authors develop an epsilon-decomposition that splits the Fisher gain into a baseline and a perturbation term and showed that the perturbation is upper bounded by squared gradient inner products. The authors proposed SPICE, a greedy data selection algorithm that scores a candidate example by the marginal gain as well as a gradient penalty term. Empirically, the authors demonstrated that at 10% data matches or outperforms full-data SFT and several baselines while reducing selection/training cost.

**Strengths:**

- This paper is very well written - clear and flows well from theory to an a practical algorithm inspired by the theory. The experiments seemed pretty complete as well.
- The proposed SPICE selection algorithm is simple and intuitive as well
- The experiments compared several baselines on multiple benchmarks. The gain is pretty consistent.

**Weaknesses:**

- It would be interesting if the authors can demonstrate whether the finding can be extended to larger corpus / base LMs as behaviour might change as we scale up the model size.
- It would be nice if the authors could provide us with qualitative examples to better understand what constitute examples that has low/high gradient conflict. Is there some intuition as to what they might imply to the data.

**Questions:**

n/a

---

> ### Author Response · Authors · 2025-11-20
> **Response to Reviewer rHBr**
>
> **`Weakness1:` Model scale and Corpus**
>
> **`Response1:`** We thank the reviewer for the suggestion. We address generalization across model scales in two parts:(1) Proxy Scaling (0.5B $\to$ 7B): Appendix H.5 evaluates varying proxy sizes (Qwen2-0.5B $\to$ 1.5B $\to$ 7B) for a Qwen2-7B target. We find that SPICE produces highly overlapping subsets and consistent downstream performance across this range, confirming stability as the proxy scales. (2) Target Scaling (up to 70B+): To directly address "scaling up," we conducted a new experiment fixing the 10% budget but scaling the target SFT model to 70B+. As shown below, using small Qwen2 proxies, SPICE consistently outperforms the "Full Data" baseline on Qwen2-72B and LLaMA2-70B. This demonstrates that the conflict-aware signal effectively transfers to substantially larger base LMs.
>
>
> | SFT Model  | SPICE (Proxy Model- Qwen2-0.5B) | SPICE (Proxy Model- Qwen2-7B) | Null | Full |
> |------------|---------------------------------|-------------------------------|------|------|
> | Qwen2-0.5B | 31.1                            | 30.8                          | 29.8 | 30.5 |
> | Qwen2-7B   | 67.0                            | 67.2                          | 62.0 | 65.2 |
> | Qwen2-72B  | 77.9                            | 78.0                          | 77.2 | 77.5 |
>
> | SFT Model  | SPICE (Proxy Model- LLaMA-7B) | Null | Full |
> |------------|-------------------------------|------|------|
> | LLaMA2-7B  | 24.1                          | 22.2 | 23.6 |
> | LLaMA2-70B | 50.9                          | 50.9 | 51.1 |
>
> Two main patterns emerge from our scaling study. (1) Within the same architecture family (e.g., Qwen2), using a small proxy model (0.5B or 7B) to select only 10% of the data yields performance that matches or even exceeds full-data LoRA SFT—even when the target model is scaled to 70B+. This shows that the conflict-aware selection signal learned by a small proxy transfers reliably to much larger models of the same family. Moreover, as the proxy scales from 0.5B to 7B, we observe highly consistent selected subsets and minimal performance variation, indicating strong proxy stability within the family.
>
> Regarding the suggestion of using very large models themselves as proxies: although we can perform LoRA SFT and evaluation on 70B+ models, using such models as proxies would require computing and storing per-sample gradients for the entire corpus, which incurs prohibitive memory and time costs. Our experiments show no clear benefit over small same-family proxies. SPICE is explicitly designed to avoid this overhead by relying on lightweight proxies to estimate Fisher geometry and conflicts, and the new 70B+ results empirically validate that this design remains effective at large scales.
>
> For substantially larger corpora, SPICE’s computational cost grows linearly with corpus size ∣𝐷∣ (complexity 𝑂(𝑘∣𝐷∣𝑑)). Once the corpus approaches pre-training scale, the selection cost becomes comparable to the training cost itself. In that regime, a more appropriate approach is to integrate conflict-aware Fisher criteria directly into the pre-training loop (e.g., via online conflict-aware sampling or methods such as Meta-rater [1]), which we view as an interesting direction beyond the scope of this work.
>
> [1] Meta-rater: A Multi-dimensional Data Selection Method for Pre-training Language Models, Zhuang et al., ACL 2025.

---

> ### Author Response · Authors · 2025-11-20
> **Response to Reviewer rHBr [Part 2]**
>
> **`Weakness2:` Examples of low/high gradient conflict**
>
> **`Response2:`** We appreciate the request for qualitative intuition. To illustrate how SPICE balances Fisher information and gradient conflict, we tracked 12 examples from a candidate pool (see Table below).
>
> 1. Intuition: Information–Conflict Trade-off.
> As highlighted in Rows 4 and 7, SPICE does not blindly discard high-conflict samples. These examples have strongly negative cosine scores (≈ –0.8) but are still selected because their information gain is substantial (Δₓ > 4). This demonstrates SPICE’s underlying mechanism: it acts as a dynamic filter that retains conflicting samples when they contribute sufficiently new and useful information.
>
> 2. Absence of surface-level patterns.
> We inspected topics, lengths, and instruction types and found no consistent linguistic patterns that distinguish high-conflict from low-conflict examples. This suggests that SPICE’s decisions are driven primarily by gradient-level geometry rather than superficial textual features.
>
> 3. Diversity verification.
> To further assess the selected subsets, we computed the NovelSum metric [1]. SPICE achieves high NovelSum scores (≈ 41–44), comparable to the full dataset. This indicates that conflict-aware filtering preserves broad semantic coverage while avoiding unnecessary redundancy, maintaining diverse and informative supervision.
>
> [1] Measuring Data Diversity for Instruction Tuning: A Systematic Analysis and A Reliable Metric （Yang et al., ACL 2025)

---

> > ### Author Response · Authors · 2025-11-20
> > **Table**
> >
> > | idx | Instruction (15 words)                                                                                                                                                                                                                                                  | cosine | $\Delta_x$ |
> > |-----|--------------------------------------------------------------------------------------------------------------------------------------------------------------------------------------------------------------------------------------------------------------|--------|----------|
> > | 0   | In the following list, select the antonym of the word "diligent" determined, honest, hardworking, sluggish                                                                                                                                                   | 0.72   | 5.48     |
> > | 1   | Classify the sentence according to its part of speech. The dog barked loudly.                                                                                                                                                                                | 0.76   | 5.02     |
> > | 2   | How can I rearrange the sentence "book reading is my favorite hobby" in a grammatically correct way? [Latex]                                                                                                                                                 | 0.83   | 4.84     |
> > | 3   | Rewrite this SQL query to select the top three records from a table named 'sales'.   Input: SELECT * FROM sales                                                                                                                                              | 0.85   | 4.80     |
> > | 4   | Group the following items based on given criteria. Apple, Banana, Orange, Carrot Criteria: Types of Fruit                                                                                                                                                    | -0.84  | 4.58     |
> > | 5   | Can you come up with a phrase that exemplifies the act of persisting through a challenging situation, using the word "persist"?                                                                                                                              | 0.81   | 4.55     |
> > | 6   | Construct a query in SQL to select users from the "Users" table aged between 30 and 40.   Input: // Table Name Users                                                                                                                                         | 0.82   | 4.41     |
> > | 7   | Generate SQL code to query a database table containing student names and ages.   Input: SELECT all students who are between 18 and 25 years of age.                                                                                                          | -0.82  | 4.18     |
> > | 8   | Using only the letters in the word "combustion", create a list of six unique words related to engines and combustion processes.                                                                                                                              | 0.82   | 4.14     |
> > | 9   | How can we optimize the performance of the given code?   Input: myList = [1, 2, 3] for item in myList:     print(item)                                                                                                                                       | 0.82   | 4.09     |
> > | 10  | Construct a for loop to access each element of the given array.   Input: [1, 2, 3, 4]                                                                                                                                                                        | 0.91   | 3.89     |
> > | 11  | Please solve the following math problem step by step:   Harly's animal shelter has 80 dogs. She adopts out 40% of them but then has to take back 5 because of personality conflicts with other dogs in their adopted homes. How many dogs does she have now? | 0.37   | 3.78     |
> > | 12  | Perform an in-place reversal of a linked list                                                                                                                                                                                                                | 0.86   | 3.74     |

---

### Official Review · Reviewer_Y2FZ · 2025-11-02

**Soundness:** 3
**Presentation:** 3
**Contribution:** 3
**Rating:** 4
**Confidence:** 3

**Summary:**

The paper proposes SPICE, a conflict-aware data selection method for instruction tuning. It starts from the submodular log-det(Fisher) objective, shows that marginal information gains decay faster when gradient conflicts are high, and formalizes this using an ε-decomposition → curvature analysis. SPICE scores each sample by (Fisher marginal) − λ·conflict (conflict = negative cosine to the running mean gradient), supports early stopping, and allows proxy models for efficiency. On LLaMA2-7B and Qwen2-7B, using ~10% of data, SPICE matches or exceeds full-data and several selectors (LESS, Fisher, SelectIT, IFD) across 8 benchmarks.

**Strengths:**

Clear theory–practice link: ε-decomposition → curvature explains greedy degradation under gradient conflicts.

Simple, practical selector: Fisher-marginal − λ·conflict with early stopping and proxy models; easy to drop into pipelines.

Solid empirical sweep: two bases (Qwen2-7B, LLaMA2-7B), 8 benchmarks, cost/ablation studies; strong gains on IFEval/MMLU at ~10% data.

**Weaknesses:**

Assumption fragility: bounds rely on α‖F‖<1 and AdaFisher approximations; reported violation rates in higher-conflict regimes weaken guarantees.

Limited baselines: several strong recent selectors and tiny-data LoRA baselines are missing; Random is competitive in places.

Conflict proxy is heuristic (−cosine to mean gradient); sensitivity to optimizer/state/batch is underexplored, and cross-architecture transfer is weak.

**Questions:**

Benchmark coverage. Please add stronger baselines, e.g., the method in arXiv:2402.02318 and a LoRA-only tiny-data baseline (e.g., 0.5–2% data) to show SPICE’s advantage at very small budgets.

Tiny-data behavior. How does SPICE compare to straightforward LoRA with small training sets across tasks? Any regime where plain LoRA beats SPICE-selected 5–10%?

---

> ### Author Response · Authors · 2025-11-20
> **Response to Reviewer Y2FZ**
>
> **`Weakness1:` Assumption fragility ($\alpha \|F\| < 1$ and AdaFisher)**
>
> **`Response1:`** We thank the reviewer for the opportunity to clarify. The condition $\alpha \|F_S\| < 1$ is a technical assumption used **only** in Appendix E to derive the closed-form upper bound in Theorem 4 (via Neumann expansion, row 210). It is **not** required for submodularity, the $\varepsilon$-decomposition, or the curvature-conflict relationship, all of which hold for any PSD Fisher matrix regardless of norm.
>
> Empirically (Appendix F.4), this condition holds in $\approx$ 90% of steps. Even in the remaining $\approx$ 10%, the bound deviation is <4.5%, indicating that the qualitative insight remains robust. Regarding **AdaFisher**, since $F^{\text{Ada}}_S$ is PSD (Appendix B), all our theoretical results apply directly to the objective optimized by SPICE. We will explicitly clarify this scope in the revision.
>
> **`Weakness3-1:` Sensitivity of optimizer, state, and batch size**
>
> **`Response2:`** In SPICE, gradients are computed at a fixed proxy checkpoint; thus, selection depends on model parameters, not directly on the optimizer state or momentum. The optimizer affects SPICE only implicitly via the checkpoint trajectory.
>
> To validate robustness, we tested varying the proxy optimizer, batch size, and seed (see Tables below).
>
> 1. **Optimizer Stability:** Changing optimizers (AdamW var., Adafactor, SGD) yields high subset overlap (Jaccard 0.84–1.00).
> 2. **Performance Stability:** While batch size and seed variations reduce overlap (0.67–0.78), downstream performance remains robust ( $\approx$ 67.0).
>
>   This confirms SPICE is **practically insensitive** to these optimization choices.
>
> | Proxy optimizer          | Jaccard overlap (selected data) |
> |--------------------------|----------------------------------|
> | AdamW (base config)      | 1.00                             |
> | AdamW ($lr \uparrow$, $\beta_2 \uparrow$)         | 0.95                             |
> | Adafactor                | 0.90                             |
> | SGD+momentum             | 0.84                             |
>
> | Batch Size | Random Seed | Jaccard overlap (selected data) | Performance (Qwen2-7B)|
> |-----------|-------------|----------------------------------|-----------------|
> | 128       | 2026        | 1.00                             | 67.0            |
> | 64        | 2026        | 0.78                             | 66.9            |
> | 256       | 2026        | 0.67                             | 67.0            |
> | 128       | 9999        | 0.77                             | 67.1            |
>
> **`Weakness3-2:` Cross-architecture transfer**
>
> **`Response3:`** Thanks for the question. SPICE is designed for **within-family proxy selection** (e.g., Qwen2-0.5B/1.5B selecting for Qwen2-7B), which is the setting targeted by our theory and main experiments. In this intended regime, different Qwen2 proxies produce highly overlapping subsets (Jaccard 0.77–1.00) and nearly identical downstream performance, indicating that small same-family proxies are stable and effective for SPICE.
>
> | Proxy Model | Qwen2-0.5B | Qwen2-1.5B | Qwen2-7B | LLaMA2-7B |
> | ----------- | ---------- | ---------- | -------- | --------- |
> | Qwen2-0.5B  | 1.00       | 0.91       | 0.77     | 0.41      |
>
> Using LLaMA2-7B as a proxy is a *cross-architecture stress test*. As the reviewer observed, the overlap drops substantially (0.41), showing that gradient geometry shifts across architectures, which naturally weakens transfer. This phenomenon is also reported by SVP, S2L, LESS, and SelectIT **[1–4]**, all of which note that proxy quality and architecture matching are key for gradient-based data selection.
>
> We will clarify that SPICE focuses on small same-family proxies, where transfer is robust and stable.
> Importantly, even under this cross-architecture stress test, SPICE still outperforms the full-data baseline, indicating that this behavior is expected rather than a weakness of the method.
>
>
>
> [1] Selection via Proxy: Efficient Data Selection for Deep Learning（Coleman et al., ICLR 2020)
>
> [2] SmallToLarge (S2L): Scalable Data Selection for Fine-tuning Large Language Models by Summarizing Training Trajectories of Small Models（Yang et al., NeurIPS 2024）
>
> [3] LESS: Selecting Influential Data for Targeted Instruction Tuning (Xia et al., ICML 2024)
>
> [4] SelectIT: Selective Instruction Tuning for LLMs via Uncertainty-aware Self-reflection（Liu et al., NeurIPS 2024）

---

> ### Author Response · Authors · 2025-11-20
> **Response to Reviewer Y2FZ [Part 2]**
>
> **`Question1&2 Weakness2:` Tiny-data (1% budget) and baselines**
>
> **`Response4:`** Thanks for pointing us to DPP, which is indeed  related to our work. In addition to the 5%, 10%, ~30%, and 100% budgets in Appendix H.7, we now more explicitly evaluate SPICE at a very small budget of 1% on both Qwen2-7B and LLaMA2-7B. And we also implemented the gradient-based DPP method and evaluated SPICE at 1–10% budgets. Across both architectures, we observe a consistent pattern:
>
> - At 1%, SPICE matches or slightly exceeds DPP/Fisher.
> - At 5–10%, SPICE consistently outperforms DPP and Fisher on average.
> - At 10%, SPICE even matches or surpasses the full-data LoRA model (58.0 vs. 56.4 on Qwen2-7B; 31.3 vs. 30.8 on LLaMA2-7B), whereas DPP/Fisher do not.
>
> We find no regime where DPP/Fisher/random on a small subset outperform SPICE; differences are within expected run-to-run variance.
>
> These results confirm that SPICE maintains a **clear advantage** in the 1–30% data budget. We will (1) add the 1% results and the DPP baseline to Appendix H.7, (2) briefly summarize in the main text that SPICE maintains an advantage in the 1–10% tiny-data regime, (3) include this DPP baseline in the main table 1&6 (row 380&1620) and (4) explicitly discuss the relation between SPICE and DPP-based selection in the related work section.
>
>
> | Qwen2-7B  | Method  | GSM8K | BBH  | MMLU | ARC-C | T-QA | IFEval | H-Eval | MBPP | AVG  |
> |-----------|---------|-------|------|------|-------|------|--------|--------|------|------|
> | 0%         | Null    | 77.8  | 60.3 | 64.4 | 48.3  | 54.3 | 25.8   | 43.9   | 53.8 | 53.6 |
> | 1%         | DPP     | 77.9  | 60.4 | 64.3 | 48.4  | 54.2 | 26.0   | 44.8   | 53.8 | 53.5 |
> | 1%         | Fisher  | 77.8  | 60.4 | 64.2 | 48.5  | 54.1 | 26.3   | 45.1   | 54.0 | 53.8 |
> | 1%         | SPICE   | 78.1  | 60.3 | 64.8 | 49.2  | 54.1 | 26.3   | 46.9   | 54.0 | 54.2 |
> | 5%         | SPICE   | 83.5  | 60.4 | 64.8 | 50.5  | 55.2 | 35.5   | 46.7   | 55.0 | 56.5 |
> | 10%        | DPP     | 86.5  | 61.0 | 66.0 | 51.0  | 55.0 | 35.4   | 45.0   | 55.7 | 57.0 |
> | 10%        | Fisher  | 86.5  | 60.8 | 65.2 | 50.3  | 55.0 | 30.6   | 44.5   | 54.6 | 55.9 |
> | 10%        | SPICE   | 86.7  | 61.0 | 67.1 | 51.8  | 55.5 | 38.6   | 47.1   | 56.2 | 58.0 |
> | ~30%       | SPICE+  | 86.7  | 61.3 | 67.3 | 52.0  | 55.5 | 38.6   | 47.5   | 56.2 | 58.1 |
> | 100%       | Full    | 84.2  | 61.3 | 65.7 | 50.5  | 54.8 | 33.5   | 45.7   | 55.2 | 56.4 |
>
> | LLaMA2-7B | Method  | GSM8K | BBH  | MMLU | ARC-C | T-QA | IFEval | H-Eval | MBPP | AVG  |
> |-----------|---------|-------|------|------|-------|------|--------|--------|------|------|
> | 0%         | Null    | 12.7  | 39.9 | 39.9 | 43.1  | 38.8 | 18.8   | 14.0   | 23.0 | 28.8 |
> | 1%         | DPP     | 12.7  | 40.0 | 40.0 | 44.0  | 38.9 | 19.1   | 14.5   | 23.0 | 28.9 |
> | 1%         | Fisher  | 13.0  | 40.0 | 40.3 | 43.5  | 38.8 | 19.0   | 14.4   | 22.8 | 29.0 |
> | 1%         | SPICE   | 12.7  | 40.1 | 40.1 | 45.9  | 38.9 | 20.0   | 14.6   | 22.4 | 29.3 |
> | 5%         | SPICE   | 12.8  | 40.5 | 40.5 | 46.0  | 40.5 | 21.0   | 16.0   | 24.0 | 30.2 |
> | 10%        | DPP     | 13.6  | 40.2 | 41.0 | 47.0  | 40.0 | 22.2   | 15.9   | 23.5 | 30.4 |
> | 10%        | Fisher  | 13.6  | 40.5 | 41.5 | 46.7  | 39.3 | 22.0   | 15.2   | 23.2 | 30.3 |
> | 10%        | SPICE   | 13.8  | 40.8 | 41.9 | 47.7  | 40.3 | 22.6   | 16.7   | 24.6 | 31.1 |
> | ~30%       | SPICE+  | 13.8  | 41.5 | 42.1 | 47.3  | 40.3 | 22.6   | 16.9   | 24.6 | 31.2 |
> | 100%       | Full    | 13.6  | 41.3 | 40.8 | 46.1  | 43.5 | 19.4   | 16.5   | 25.2 | 30.8 |
>
>
>
> **`Weakness3-3:` Conflict proxy**
>
> **`Response5:`** We thank the reviewer for raising this comment. Our theoretical analysis is formulated in terms of gradient inner-product interactions, in particular through the perturbation term $|\varepsilon_x(S)|$ that depends on $\sum |g_x^T g_y|^2$. The practical conflict score $-cos(g_x,\bar{g})$ is a scale-invariant and computationally efficient instantiation of this idea.
>
> Empirically, this proxy is tightly linked to the $\varepsilon_x$ term we study: it exhibits a strong correlation with (Spearman $\rho=$ 0.901), indicating that it accurately tracks the perturbation governing marginal Fisher gains. Moreover, in our ablation, removing the conflict penalty (i.e., no conflict term) consistently degrades downstream performance. Taken together, these results show that our conflict score is an **efficient and effective approximation** to the underlying inner-product interactions predicted by our theory, **rather than an arbitrary heuristic**.

---

> > ### Comment · Reviewer_Y2FZ · 2025-11-20
> >
> > thanks for authors response. the paper is an increamental innovation from the prior works but i do like the idea of using proxy model for the selection to be more efficent. Though the results are not overwhleming and it's a question that whether this can scale to today's foundation model with size over 100B and context window over 1M. Given those considerations i raise my score to 6.

---

> > > ### Author Response · Authors · 2025-11-24
> > > **Response to Reviewer Y2FZ**
> > >
> > > We sincerely thank the reviewer for the constructive feedback and for raising the score. We appreciate the reviewer's recognition of our contribution.
> > >
> > > Regarding the scalability to 100B+ models and million-token context windows, we note that SPICE itself is agnostic to the target model size and maximum context length, as the selection criterion depends only on gradient geometry and is computed entirely on a lightweight proxy model.
> > >
> > > Across the 0.5B-70B scale, our experiments already show that this gradient geometry transfers reliably: subsets selected by small proxy models achieve performance that matches or exceeds full-data SFT on 7B and 70B targets.
> > >
> > > For ultra-long inputs, the practical limitation comes from the proxy model's input window rather than from the SPICE method itself. In current instruction-tuning corpora, nearly all samples are far shorter than the proxy window, so this does not affect our reported results. As larger long-context checkpoints and more efficient gradient extraction techniques (e.g., chunked or projected gradients) become available, we plan to extend SPICE to such settings.
> > >
> > > We thank the reviewer again for highlighting this important direction and for their positive assessment of our work.

---

### Author Response · Authors · 2025-11-27
**General Response**

Dear AC and Reviewers,

We would like to sincerely thank you for taking the time to handle our submission, especially given the recent adjustments to the ICLR review process. We are also grateful to all four reviewers for their detailed and constructive feedback.

Across all four reviews, concerns mainly focus on (i) theoretical assumption clarity, (ii) tiny-data baselines and the 1% regime, (iii) scalability to larger (70B+) models, (iv) diversity–conflict trade-offs, (v) computational cost, and (vi) proxy-to-target transfer.

We addressed each point with new experiments, ablations, and clarifications.
In particular:

- **Scalability.** We show that SPICE-selected subsets from small same-family proxies transfer reliably to 70B+ models, matching the full-data baseline.
- **Effectiveness.** With 10% data, SPICE consistently matches or exceeds full-data LoRA across both Qwen2-7B and LLaMA2-7B, including new added baselines.
- **Tiny-data regime.** At 1–10%, SPICE outperforms Fisher, DPP, LESS, IFD, SelectIT, TSDS, LEAD, and random.
- **Diversity.** NovelSum/LDD and domain-coverage results show that conflict-aware filtering preserves semantic diversity and reduces harmful conflicts.
- **Efficiency.** SPICE has a single-pass, linear-time complexity in $|D|$ and proxy size, and does not require recomputing gradients.

Current state of the reviews after the revision:
- **Reviewer Y2FZ** has raised their score to **6 (weak accept)** after considering the new tiny-data experiments, additional baselines, and proxy-scaling results, and explicitly notes that they like the efficiency of using proxy models for selection.

- **Reviewer rHBr** keeps a **6 (weak accept)** and highlights the clear theory–to–algorithm flow and consistent empirical gains. The additional scaling and qualitative analyses in the revision further support these strengths.

- **Reviewer Xhgm** currently gives a **6 (weak accept)**; after the additional experiments and clarifications, their remaining comment focuses only on improving the clarity and ordering of the theoretical exposition around Theorem 1. The latest revised version already incorporates this restructuring, so the paper now reflects their requested changes and further strengthens their already positive assessment.

- **Reviewer KYWw** consistently gives an **8 (accept, good paper/poster)**, emphasizing the practical value of combining effectiveness and efficiency and the breadth of the experimental study.  The new proxy-sensitivity, diversity, and cost analyses in the revision directly address the questions raised in their review.

All of these new results and clarifications have been incorporated into the revised manuscript and documented in our point-by-point summary of revisions.

We sincerely thank all reviewers for the thoughtful feedback, which significantly improved the clarity and completeness of the work.

Sincerely,

The Authors

---

### Author Response · Authors · 2025-11-27
**Summary of Revisions**

Dear AC and Reviewers,

We thanks for your thoughtful and constructive feedback.  In the revised version, we have made a number of changes to address the main concerns. A brief summary of the key revisions is provided below. All new or modified text is highlighted in **green** in the updated manuscript.

- **Conceptual / theoretical clarifications**
  - **Section 2.2 (rows 167–168):** We refine the description of $\varepsilon_x(S)$ and explicitly state that $|\varepsilon_x(S)|$ measures the cumulative decay of the marginal gain from $\Delta_x(\varnothing)$ to $\Delta_x(S)$.
  - **Section 2.3 (rows 203-205):** We add a concise sentence expressing the curvature numerator in terms of $\varepsilon_x (D \setminus {x})$.
  - **Section 2.3 (rows 214–215):** We clarify the scope and empirical validity of the technical condition $\alpha\|F_S\| < 1$ and emphasize that it is used only in the Neumann-series bound (Theorem 4).
  - **Section 4.3 (rows 346–347):** We clarify that the phrase “at each iteration, we select one sample” refers to a greedy step within a fixed candidate pool, rather than an online per-example update.

- **Baseline extensions (DPP, TSDS, LEAD)**
  - **Section 5.1 (rows 369–370):** We include the new baselines in the experimental setup description.
  - **Table 1 (rows 385–399):** We add the results of the new baselines on the 8 benchmarks.
  - **Table 2 (rows 432–446):** We move the original Table 6 (rows 1620–1634) into the main text as Table 2 and incorporate the new baselines’ results, together with a corresponding explanation in Section 5.2 (rows 414–416).
  - **Figure 3 (rows 452–461):** We include cost analysis results for the new baselines.

- **Model scale and tiny-data regime**
  - **Model scale:** We add results for fine-tuning 70B+ models in Appendix H.4 (rows 1779–1808) and summarize the corresponding findings in Section 5.2 (rows 416–417).
  - **Tiny data:** In addition to the existing experiments with 5% of the data, we add experiments using 1% of the data in Appendix H.6 (Table 14), and reference these results in the main text (row 418).

- **Cost analysis**
  - **Section 5.3 (rows 430–431):** We provide a more explicit discussion of computational complexity.
  - **Appendix H.7 and H.8 (rows 1933–2026):** We give a more detailed explanation of both computational and memory complexity, and report the concrete selection time.

- **Diversity analysis**
  - **Section 5.5 (rows 491–507):** We add experiments analyzing the diversity of the selected data subsets, showing that our method is able to select diverse subsets while maintaining high performance.

- **Related work**
  - **Section 6 (rows 516–523):** We incorporate additional related work referenced by the reviewers [1-6].

We sincerely appreciate the reviewers’ comments and suggestions, which have helped us improve the clarity, completeness, and empirical support of the paper. We hope that the revised manuscript adequately addresses your concerns.

Sincerely,

The Authors

---
[1] Ivison, H., Zhang, M., Brahman, F., Koh, P. W., & Dasigi, P. (2025). Large-Scale Data Selection for Instruction Tuning. arXiv preprint arXiv:2503.01807.

[2] Liu, Z., Karbasi, A., & Rekatsinas, T. (2024). TSDS: Data Selection for Task-Specific Model Finetuning. In The Thirty-eighth Annual Conference on Neural Information Processing Systems (NeurIPS 2024).

[3] Wang, J., Lin, X., Qiao, R., Koh, P. W., Foo, C.-S., & Low, B. K. H. (2025). NICE Data Selection for Instruction Tuning in LLMs with Non-differentiable Evaluation Metric. In Forty-second International Conference on Machine Learning (ICML 2025).

[4] Chen, Y., Li, Y., Hu, K., Ma, Z., Ye, H., & Chen, K. (2025). MIG: Automatic Data Selection for Instruction Tuning by Maximizing Information Gain in Semantic Space. In Findings of the Association for Computational Linguistics: ACL 2025.

[5] Wang, P., Shen, Y., Guo, Z., Stallone, M., Kim, Y., Golland, P., & Panda, R. (2024). Diversity Measurement and Subset Selection for Instruction Tuning Datasets. arXiv.

[6] Yang, Y., Nan, Y., Ye, J., Dou, S., Wang, X., Li, S., Lv, H., Gui, T., Zhang, Q., & Huang, X. (2025). Measuring Data Diversity for Instruction Tuning: A Systematic Analysis and A Reliable Metric. In Proceedings of the 63rd Annual Meeting of the Association for Computational Linguistics Association for Computational Linguistics

---

### Meta-Review · Area_Chair_u4hM · 2026-01-08

**Summary:**

The reviewers initially expressed several concerns regarding the theoretical foundations, empirical rigor, and practical scalability of the proposed SPICE method:

- Theoretical Clarity and Assumptions: Reviewers Y2FZ and Xhgm questioned the fragility of the theoretical assumptions, the logical flow of the derivation explaining why large perturbations lead to faster marginal gain decay.

- Baselines and Tiny-Data Regime: Reviewers Y2FZ and Xhgm noted missing baselines and requested comparisons at very small data budgets to verify the method's effectiveness in extreme low-resource settings.

- Scalability and Transferability: Reviewers rHBr and KYWw raised concerns about whether the method, tested primarily on 7B models, would scale to larger models (70B+) and how sensitive the selection is to the choice of proxy model.

- Computational Cost: Reviewers Xhgm and KYWw requested more granular breakdowns of computational complexity and wall-clock time to justify the "efficiency" claims compared to online/iterative methods.

- Diversity vs. Conflict: Reviewer KYWw worried that penalizing gradient conflicts might suppress useful diversity in the data.

**Reviewer Concerns:**

Addressed Concerns:

Baselines & Tiny Data: The authors added extensive experiments with new baselines and evaluations at small data budgets. The results demonstrated SPICE's consistent superiority, satisfying Reviewers Y2FZ and Xhgm.

Scalability: The authors provided new results showing that data selected by small proxies transfers effectively to large target models, matching full-data performance. This addressed the scalability concerns of rHBr and KYWw.

Computational Cost: The authors clarified the streaming nature of the algorithm and provided detailed time cost tables showing SPICE is faster than full-data training and competitive with other selection methods.

Diversity: The authors provided diversity metrics and domain coverage analysis, showing that conflict-aware selection preserves semantic diversity, addressing KYWw's concern.

Theoretical Clarifications: The authors clarified that the strict norm assumption is only required for one specific bound, not the general framework.

Outstanding Concerns:

Reviewer Xhgm noted that while the theoretical explanations were improved during the rebuttal, the specific logical ordering of the definitions and theorems in the revised PDF was still slightly disjointed. The authors have committed to a specific re-ordering plan for the final version to resolve this coherence issue.

The authors acknowledged that cross-architecture transfer is weak. This is a known limitation of gradient-based selection methods in general, rather than a specific flaw of this paper, but remains a constraint of the approach.

**Reviewer Scores:**

All reviewers participated in the discussion or acknowledged the rebuttal.

Reviewer Y2FZ: Raised score from 4 to 6 (only reflected in the reviewer's comment). The reviewer was satisfied with the additional experiments in the tiny-data regime and the inclusion of missing baselines, noting the method's efficiency using proxy models is a strong point.

Reviewer Xhgm: Maintained score at 6. The reviewer acknowledged that most concerns regarding baselines and complexity were resolved. They kept the score at Weak Accept primarily due to the presentation structure of the theory section, which they expect to be fixed in the final version based on the authors' proposal.

Reviewer rHBr: Maintained score at 6. The reviewer remained positive about the clear theory-to-practice flow and was satisfied with the additional scalability results and qualitative examples provided during the rebuttal.

Reviewer KYWw: Maintained score at 8, though didn't respond to the rebuttal. The reviewer strongly supported the paper throughout, emphasizing the practical value of the efficiency/effectiveness trade-off. The additional sensitivity and diversity analyses strengthened their positive assessment.

---

### Decision · Program_Chairs · 2026-01-26

Accept (Poster)